# Fisher GAN

**Youssef Mroueh**∗**, Tom Sercu**∗
mroueh@us.ibm.com, tom.sercu1@ibm.com
∗ Equal Contribution
AI Foundations, IBM Research AI
IBM T.J Watson Research Center

## Abstract

Generative Adversarial Networks (GANs) are powerful models for learning complex distributions. Stable training of GANs has been addressed in many recent works which explore different metrics between distributions. In this paper we introduce *Fisher GAN* which fits within the Integral Probability Metrics (IPM) framework for training GANs. Fisher GAN defines a critic with a data dependent constraint on its *second order moments*. We show in this paper that Fisher GAN allows for stable and time efficient training that does not compromise the capacity of the critic, and does not need data independent constraints such as weight clipping. We analyze our Fisher IPM theoretically and provide an algorithm based on Augmented Lagrangian for Fisher GAN. We validate our claims on both image sample generation and semi-supervised classification using Fisher GAN.

## 1   Introduction

Generative Adversarial Networks (GANs) [1] have recently become a prominent method to learn high-dimensional probability distributions. The basic framework consists of a generator neural network which learns to generate samples which approximate the distribution, while the discriminator measures the distance between the real data distribution, and this learned distribution that is referred to as *fake* distribution. The generator uses the gradients from the discriminator to minimize the distance with the real data distribution. The distance between these distributions was the object of study in [2], and highlighted the impact of the distance choice on the stability of the optimization. The original GAN formulation optimizes the Jensen-Shannon divergence, while later work generalized this to optimize f-divergences [3], KL [4], the Least Squares objective [5]. Closely related to our work, Wasserstein GAN (WGAN) [6] uses the earth mover distance, for which the discriminator function class needs to be constrained to be Lipschitz. To impose this Lipschitz constraint, WGAN proposes to use *weight clipping*, i.e. a data independent constraint, but this comes at the cost of reducing the capacity of the critic and high sensitivity to the choice of the clipping hyper-parameter. A recent development Improved Wasserstein GAN (WGAN-GP) [7] introduced a data dependent constraint namely a *gradient penalty* to enforce the Lipschitz constraint on the critic, which does not compromise the capacity of the critic but comes at a high computational cost.

We build in this work on the Integral probability Metrics (IPM) framework for learning GAN of [8]. Intuitively the IPM defines a *critic* function $f$, that maximally discriminates between the real and fake distributions. We propose a theoretically sound and time efficient data dependent constraint on the critic of Wasserstein GAN, that allows a stable training of GAN and does not compromise the capacity of the critic. Where WGAN-GP uses a penalty on the gradients of the critic, *Fisher GAN* imposes a constraint on the *second order moments* of the critic. This extension to the IPM framework is inspired by the *Fisher Discriminant Analysis* method.

The main contributions of our paper are:

1. We introduce in Section 2 the Fisher IPM, a scaling invariant distance between distributions. Fisher IPM introduces a data dependent constraint on the second order moments of the critic that discriminates between the two distributions. Such a constraint ensures the boundedness of the metric and the critic. We show in Section 2.2 that Fisher IPM when approximated with neural networks, corresponds to a discrepancy between *whitened* mean feature embeddings of the distributions. In other words a mean feature discrepancy that is measured with a Mahalanobis distance in the space computed by the neural network.

2. We show in Section 3 that Fisher IPM corresponds to the *Chi-squared* distance ($\chi_2$) when the critic has unlimited capacity (the critic belongs to a universal hypothesis function class). Moreover we prove in Theorem 2 that even when the critic is parametrized by a neural network, it approximates the $\chi_2$ distance with a factor which is a inner product between optimal and neural network critic. We finally derive generalization bounds of the learned critic from samples from the two distributions, assessing the statistical error and its convergence to the Chi-squared distance from finite sample size.

3. We use Fisher IPM as a GAN objective [1] and formulate an algorithm that combines desirable properties (Table 1): a stable and meaningful loss between distributions for GAN as in Wasserstein GAN [6], at a low computational cost similar to simple weight clipping, while not compromising the capacity of the critic via a data dependent constraint but at a much lower computational cost than [7]. Fisher GAN achieves strong semi-supervised learning results without need of batch normalization in the critic.

Table 1: Comparison between Fisher GAN and recent related approaches.

|  | Stability | Unconstrained capacity | Efficient Computation | Representation power (SSL) |
|---|---|---|---|---|
| Standard GAN [1, 9] | ✗ | ✓ | ✓ | ✓ |
| WGAN, McGan [6, 8] | ✓ | ✗ | ✓ | ✗ |
| WGAN-GP [7] | ✓ | ✓ | ✗ | ? |
| Fisher Gan (Ours) | ✓ | ✓ | ✓ | ✓ |

## 2 Learning GANs with Fisher IPM

### 2.1 Fisher IPM in an arbitrary function space: General framework

**Integral Probability Metric (IPM).** Intuitively an IPM defines a critic function $f$ belonging to a function class $\mathscr{F}$, that maximally discriminates between two distributions. The function class $\mathscr{F}$ defines how $f$ is bounded, which is crucial to define the metric. More formally, consider a compact space $\mathcal{X}$ in $\mathbb{R}^d$. Let $\mathscr{F}$ be a set of *measurable, symmetric and bounded* real valued functions on $\mathcal{X}$. Let $\mathscr{P}(\mathcal{X})$ be the set of measurable probability distributions on $\mathcal{X}$. Given two probability distributions $\mathbb{P}, \mathbb{Q} \in \mathscr{P}(\mathcal{X})$, the IPM indexed by a *symmetric* function space $\mathscr{F}$ is defined as follows [10]:

$$d_{\mathscr{F}}(\mathbb{P}, \mathbb{Q}) = \sup_{f \in \mathscr{F}} \left\{ \mathbb{E}_{x \sim \mathbb{P}} f(x) - \mathbb{E}_{x \sim \mathbb{Q}} f(x) \right\}. \tag{1}$$

It is easy to see that $d_{\mathscr{F}}$ defines a pseudo-metric over $\mathscr{P}(\mathcal{X})$. Note specifically that if $\mathscr{F}$ is not bounded, $\sup_f$ will scale $f$ to be arbitrarily large. By choosing $\mathscr{F}$ appropriately [11], various distances between probability measures can be defined.

**First formulation: Rayleigh Quotient.** In order to define an IPM in the GAN context, [6, 8] impose the boundedness of the function space via a data independent constraint. This was achieved via restricting the norms of the weights parametrizing the function space to a $\ell_p$ ball. Imposing such a data independent constraint makes the training highly dependent on the constraint hyper-parameters and restricts the capacity of the learned network, limiting the usability of the learned critic in a semi-supervised learning task. Here we take a different angle and design the IPM to be scaling invariant as a *Rayleigh quotient*. Instead of measuring the discrepancy between means as in Equation (1), we measure a *standardized* discrepancy, so that the distance is **bounded by construction**. Standardizing this discrepancy introduces as we will see a *data dependent* constraint, that controls the growth of the weights of the critic $f$ and ensures the stability of the training while maintaining the capacity of the critic. Given two distributions $\mathbb{P}, \mathbb{Q} \in \mathscr{P}(\mathcal{X})$ the Fisher IPM for a function space $\mathscr{F}$ is defined as follows:

$$d_{\mathscr{F}}(\mathbb{P}, \mathbb{Q}) = \sup_{f \in \mathscr{F}} \frac{\mathbb{E}_{x \sim \mathbb{P}}[f(x)] - \mathbb{E}_{x \sim \mathbb{Q}}[f(x)]}{\sqrt{1/2 \mathbb{E}_{x \sim \mathbb{P}} f^2(x) + 1/2 \mathbb{E}_{x \sim \mathbb{Q}} f^2(x)}}. \tag{2}$$

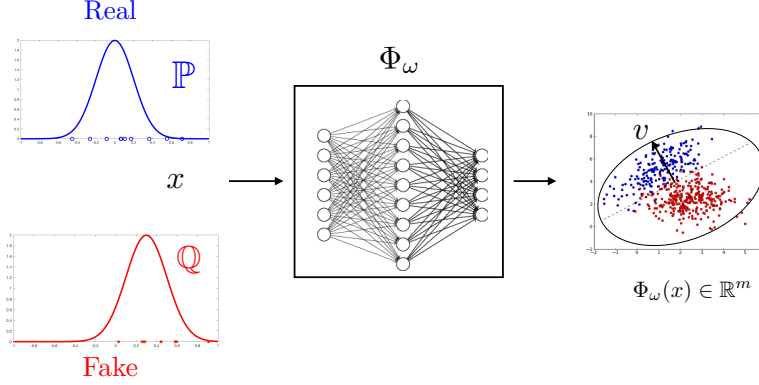

Figure 1: Illustration of Fisher IPM with Neural Networks. $\Phi_\omega$ is a convolutional neural network which defines the embedding space. $v$ is the direction in this embedding space with maximal mean separation $\langle v, \mu_\omega(\mathbb{P}) - \mu_\omega(\mathbb{Q}) \rangle$, constrained by the hyperellipsoid $v^\top \Sigma_\omega(\mathbb{P}; \mathbb{Q}) v = 1$.

While a standard IPM (Equation (1)) maximizes the discrepancy between the means of a function under two different distributions, Fisher IPM looks for critic $f$ that achieves a tradeoff between maximizing the discrepancy between the means under the two distributions (between *class* variance), and reducing the pooled second order moment (an upper bound on the intra-class variance).

Standardized discrepancies have a long history in statistics and the so-called *two-samples* hypothesis testing. For example the classic two samples Student's $t-$ test defines the student statistics as the ratio between means discrepancy and the sum of standard deviations. It is now well established that learning generative models has its roots in the two-samples hypothesis testing problem [12]. Non parametric two samples testing and model criticism from the kernel literature lead to the so called maximum kernel mean discrepancy (MMD) [13]. The MMD cost function and the mean matching IPM for a general function space has been recently used for training GAN [14, 15, 8].

Interestingly Harchaoui et al [16] proposed Kernel Fisher Discriminant Analysis for the two samples hypothesis testing problem, and showed its statistical consistency. The Standard Fisher discrepancy used in Linear Discriminant Analysis (LDA) or Kernel Fisher Discriminant Analysis (KFDA) can be written: $\sup_{f \in \mathscr{F}} \frac{\left( \mathbb{E}_{x \sim \mathbb{P}}[f(x)] - \mathbb{E}_{x \sim \mathbb{Q}}[f(x)] \right)^2}{\text{Var}_{x \sim \mathbb{P}}(f(x)) + \text{Var}_{x \sim \mathbb{Q}}(f(x))}$, where $\text{Var}_{x \sim \mathbb{P}}(f(x)) = \mathbb{E}_{x \sim \mathbb{P}} f^2(x) - (\mathbb{E}_{x \sim \mathbb{P}}(f(x)))^2$. Note that in LDA $\mathscr{F}$ is restricted to linear functions, in KFDA $\mathscr{F}$ is restricted to a Reproducing Kernel Hilbert Space (RKHS). Our Fisher IPM (Eq (2)) deviates from the standard Fisher discrepancy since the numerator is not squared, and we use in the denominator the second order moments instead of the variances. Moreover in our definition of Fisher IPM, $\mathscr{F}$ can be any symmetric function class.

**Second formulation: Constrained form.** Since the distance is scaling invariant, $d_{\mathscr{F}}$ can be written equivalently in the following constrained form:

$$d_{\mathscr{F}}(\mathbb{P}, \mathbb{Q}) = \sup_{f \in \mathscr{F}, \frac{1}{2}\mathbb{E}_{x \sim \mathbb{P}} f^2(x) + \frac{1}{2}\mathbb{E}_{x \sim \mathbb{Q}} f^2(x) = 1} \mathcal{E}(f) := \mathbb{E}_{x \sim \mathbb{P}}[f(x)] - \mathbb{E}_{x \sim \mathbb{Q}}[f(x)]. \quad (3)$$

**Specifying $\mathbb{P}, \mathbb{Q}$: Learning GAN with Fisher IPM.** We turn now to the problem of learning GAN with Fisher IPM. Given a distribution $\mathbb{P}_r \in \mathscr{P}(\mathcal{X})$, we learn a function $g_\theta : \mathcal{Z} \subset \mathbb{R}^{n_z} \to \mathcal{X}$, such that for $z \sim p_z$, the distribution of $g_\theta(z)$ is close to the real data distribution $\mathbb{P}_r$, where $p_z$ is a fixed distribution on $\mathcal{Z}$ (for instance $z \sim \mathcal{N}(0, I_{n_z})$). Let $\mathbb{P}_\theta$ be the distribution of $g_\theta(z), z \sim p_z$. Using Fisher IPM (Equation (3)) indexed by a parametric function class $\mathscr{F}_p$, the generator minimizes the IPM: $\min_{g_\theta} d_{\mathscr{F}_p}(\mathbb{P}_r, \mathbb{P}_\theta)$. Given samples $\{x_i, 1 \ldots N\}$ from $\mathbb{P}_r$ and samples $\{z_i, 1 \ldots M\}$ from $p_z$ we shall solve the following empirical problem:

$$\min_{g_\theta} \sup_{f_p \in \mathscr{F}_p} \hat{\mathcal{E}}(f_p, g_\theta) := \frac{1}{N} \sum_{i=1}^{N} f_p(x_i) - \frac{1}{M} \sum_{j=1}^{M} f_p(g_\theta(z_j)) \text{ Subject to } \hat{\Omega}(f_p, g_\theta) = 1, \quad (4)$$

where $\hat{\Omega}(f_p, g_\theta) = \frac{1}{2N} \sum_{i=1}^{N} f_p^2(x_i) + \frac{1}{2M} \sum_{j=1}^{M} f_p^2(g_\theta(z_j))$. For simplicity we will have $M = N$.

## 2.2 Fisher IPM with Neural Networks

We will specifically study the case where $\mathscr{F}$ is a finite dimensional Hilbert space induced by a neural network $\Phi_\omega$ (see Figure 1 for an illustration). In this case, an IPM with data-independent constraint will be equivalent to mean matching [8]. We will now show that Fisher IPM will give rise to a *whitened* mean matching interpretation, or equivalently to *mean matching* with a Mahalanobis distance.

**Rayleigh Quotient.** Consider the function space $\mathscr{F}_{v,\omega}$, defined as follows

$$\mathscr{F}_{v,\omega} = \{f(x) = \langle v, \Phi_\omega(x) \rangle \,|\, v \in \mathbb{R}^m, \Phi_\omega : \mathcal{X} \to \mathbb{R}^m\},$$

$\Phi_\omega$ is typically parametrized with a multi-layer neural network. We define the mean and covariance (Gramian) feature embedding of a distribution as in McGan [8]:

$$\mu_\omega(\mathbb{P}) = \mathop{\mathbb{E}}_{x \sim \mathbb{P}}(\Phi_\omega(x)) \quad \text{and} \quad \Sigma_\omega(\mathbb{P}) = \mathop{\mathbb{E}}_{x \sim \mathbb{P}}\left(\Phi_\omega(x)\Phi_\omega(x)^\top\right),$$

Fisher IPM as defined in Equation (2) on $\mathscr{F}_{v,\omega}$ can be written as follows:

$$d_{\mathscr{F}_{v,\omega}}(\mathbb{P}, \mathbb{Q}) = \max_\omega \max_v \frac{\langle v, \mu_\omega(\mathbb{P}) - \mu_\omega(\mathbb{Q}) \rangle}{\sqrt{v^\top(\frac{1}{2}\Sigma_\omega(\mathbb{P}) + \frac{1}{2}\Sigma_\omega(\mathbb{Q}) + \gamma I_m)v}}, \tag{5}$$

where we added a regularization term ($\gamma > 0$) to avoid singularity of the covariances. Note that if $\Phi_\omega$ was implemented with homogeneous non linearities such as RELU, if we swap $(v, \omega)$ with $(cv, c'\omega)$ for any constants $c, c' > 0$, the distance $d_{\mathscr{F}_{v,\omega}}$ remains unchanged, hence the scaling invariance.

**Constrained Form.** Since the Rayleigh Quotient is not amenable to optimization, we will consider Fisher IPM as a constrained optimization problem. By virtue of the scaling invariance and the constrained form of the Fisher IPM given in Equation (3), $d_{\mathscr{F}_{v,\omega}}$ can be written equivalently as:

$$d_{\mathscr{F}_{v,\omega}}(\mathbb{P}, \mathbb{Q}) = \max_{\omega, v, v^\top(\frac{1}{2}\Sigma_\omega(\mathbb{P}) + \frac{1}{2}\Sigma_\omega(\mathbb{Q}) + \gamma I_m)v = 1} \langle v, \mu_\omega(\mathbb{P}) - \mu_\omega(\mathbb{Q}) \rangle \tag{6}$$

Define the pooled covariance: $\Sigma_\omega(\mathbb{P}; \mathbb{Q}) = \frac{1}{2}\Sigma_\omega(\mathbb{P}) + \frac{1}{2}\Sigma_\omega(\mathbb{Q}) + \gamma I_m$. Doing a simple change of variable $u = (\Sigma_\omega(\mathbb{P}; \mathbb{Q}))^{\frac{1}{2}} v$ we see that:

$$
\begin{aligned}
d_{\mathscr{F}_{u,\omega}}(\mathbb{P}, \mathbb{Q}) &= \max_\omega \max_{u, \|u\|=1} \left\langle u, (\Sigma_\omega(\mathbb{P}; \mathbb{Q}))^{-\frac{1}{2}}(\mu_\omega(\mathbb{P}) - \mu_\omega(\mathbb{Q})) \right\rangle \\
&= \max_\omega \left\| (\Sigma_\omega(\mathbb{P}; \mathbb{Q}))^{-\frac{1}{2}}(\mu_\omega(\mathbb{P}) - \mu_\omega(\mathbb{Q})) \right\|,
\end{aligned} \tag{7}
$$

hence we see that fisher IPM corresponds to the worst case distance between *whitened means*. Since the means are white, we don't need to impose further constraints on $\omega$ as in [6, 8]. Another interpretation of the Fisher IPM stems from the fact that:

$$d_{\mathscr{F}_{v,\omega}}(\mathbb{P}, \mathbb{Q}) = \max_\omega \sqrt{(\mu_\omega(\mathbb{P}) - \mu_\omega(\mathbb{Q}))^\top \Sigma_\omega^{-1}(\mathbb{P}; \mathbb{Q})(\mu_\omega(\mathbb{P}) - \mu_\omega(\mathbb{Q}))},$$

from which we see that Fisher IPM is a Mahalanobis distance between the mean feature embeddings of the distributions. The Mahalanobis distance is defined by the positive definite matrix $\Sigma_w(\mathbb{P}; \mathbb{Q})$. We show in Appendix A that the gradient penalty in Improved Wasserstein [7] gives rise to a similar Mahalanobis mean matching interpretation.

**Learning GAN with Fisher IPM.** Hence we see that learning GAN with Fisher IPM:

$$\min_{g_\theta} \max_\omega \max_{v, v^\top(\frac{1}{2}\Sigma_\omega(\mathbb{P}_r) + \frac{1}{2}\Sigma_\omega(\mathbb{P}_\theta) + \gamma I_m)v = 1} \langle v, \mu_w(\mathbb{P}_r) - \mu_\omega(\mathbb{P}_\theta) \rangle$$

corresponds to a min-max game between a feature space and a generator. The feature space tries to maximize the Mahalanobis distance between the feature means embeddings of *real* and *fake* distributions. The generator tries to minimize the mean embedding distance.

## 3 Theory

We will start first by studying the Fisher IPM defined in Equation (2) when the function space has *full capacity* i.e when the critic belongs to $\mathscr{L}_2(\mathcal{X}, \frac{1}{2}(\mathbb{P}+\mathbb{Q}))$ meaning that $\int_\mathcal{X} f^2(x) \frac{(\mathbb{P}(x) + \mathbb{Q}(x))}{2} dx < \infty$. Theorem 1 shows that under this condition, the Fisher IPM corresponds to the *Chi-squared distance* between distributions, and gives a closed form expression of the *optimal critic* function $f_\chi$ (See Appendix B for its relation with the Pearson Divergence). Proofs are given in Appendix D.

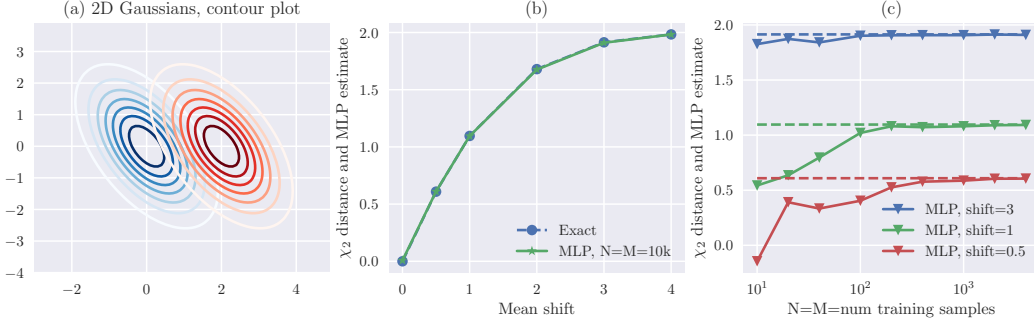

Figure 2: Example on 2D synthetic data, where both $\mathbb{P}$ and $\mathbb{Q}$ are fixed normal distributions with the same covariance and shifted means along the x-axis, see (a). Fig (b, c) show the exact $\chi_2$ distance from numerically integrating Eq (8), together with the estimate obtained from training a 5-layer MLP with layer size = 16 and LeakyReLU nonlinearity on different training sample sizes. The MLP is trained using Algorithm 1, where sampling from the generator is replaced by sampling from $\mathbb{Q}$, and the $\chi_2$ MLP estimate is computed with Equation (2) on a large number of samples (i.e. out of sample estimate). We see in (b) that for large enough sample size, the MLP estimate is extremely good. In (c) we see that for smaller sample sizes, the MLP approximation bounds the ground truth $\chi_2$ from below (see Theorem 2) and converges to the ground truth roughly as $\mathcal{O}(\frac{1}{\sqrt{N}})$ (Theorem 3). We notice that when the distributions have small $\chi_2$ distance, a larger training size is needed to get a better estimate - again this is in line with Theorem 3.

**Theorem 1** (Chi-squared distance at full capacity). *Consider the Fisher IPM for $\mathscr{F}$ being the space of all measurable functions endowed by $\frac{1}{2}(\mathbb{P} + \mathbb{Q})$, i.e. $\mathscr{F} := \mathscr{L}_2(\mathcal{X}, \frac{\mathbb{P}+\mathbb{Q}}{2})$. Define the Chi-squared distance between two distributions:*

$$\chi_2(\mathbb{P}, \mathbb{Q}) = \sqrt{\int_{\mathcal{X}} \frac{(\mathbb{P}(x) - \mathbb{Q}(x))^2}{\frac{\mathbb{P}(x)+\mathbb{Q}(x)}{2}} dx} \qquad (8)$$

*The following holds true for any $\mathbb{P}, \mathbb{Q}, \mathbb{P} \neq \mathbb{Q}$:*

*1) The Fisher IPM for $\mathscr{F} = \mathscr{L}_2(\mathcal{X}, \frac{\mathbb{P}+\mathbb{Q}}{2})$ is equal to the Chi-squared distance defined above: $d_{\mathscr{F}}(\mathbb{P}, \mathbb{Q}) = \chi_2(\mathbb{P}, \mathbb{Q})$.*
*2) The optimal critic of the Fisher IPM on $\mathscr{L}_2(\mathcal{X}, \frac{\mathbb{P}+\mathbb{Q}}{2})$ is :*

$$f_\chi(x) = \frac{1}{\chi_2(\mathbb{P}, \mathbb{Q})} \frac{\mathbb{P}(x) - \mathbb{Q}(x)}{\frac{\mathbb{P}(x)+\mathbb{Q}(x)}{2}}.$$

We note here that LSGAN [5] at full capacity corresponds to a Chi-Squared divergence, with the main difference that LSGAN has different objectives for the generator and the discriminator (bilevel optimizaton), and hence does not optimize a single objective that is a distance between distributions. The Chi-squared divergence can also be achieved in the $f$-gan framework from [3]. We discuss the advantages of the Fisher formulation in Appendix C.

Optimizing over $\mathscr{L}_2(\mathcal{X}, \frac{\mathbb{P}+\mathbb{Q}}{2})$ is not tractable, hence we have to restrict our function class, to a hypothesis class $\mathscr{H}$, that enables tractable computations. Here are some typical choices of the space $\mathscr{H}$ : Linear functions in the input features, RKHS, a non linear multilayer neural network with a linear last layer ($\mathscr{F}_{v,\omega}$). In this Section we don't make any assumptions about the function space and show in Theorem 2 how the Chi-squared distance is approximated in $\mathscr{H}$, and how this depends on the approximation error of the optimal critic $f_\chi$ in $\mathscr{H}$.

**Theorem 2** (Approximating Chi-squared distance in an arbitrary function space $\mathscr{H}$). *Let $\mathscr{H}$ be an arbitrary symmetric function space. We define the inner product $\langle f, f_\chi \rangle_{\mathscr{L}_2(\mathcal{X}, \frac{\mathbb{P}+\mathbb{Q}}{2})} = \int_{\mathcal{X}} f(x) f_\chi(x) \frac{\mathbb{P}(x)+\mathbb{Q}(x)}{2} dx$, which induces the Lebesgue norm. Let $\mathbb{S}_{\mathscr{L}_2(\mathcal{X}, \frac{\mathbb{P}+\mathbb{Q}}{2})}$ be the unit sphere in $\mathscr{L}_2(\mathcal{X}, \frac{\mathbb{P}+\mathbb{Q}}{2})$: $\mathbb{S}_{\mathscr{L}_2(\mathcal{X}, \frac{\mathbb{P}+\mathbb{Q}}{2})} = \{f : \mathcal{X} \to \mathbb{R}, \|f\|_{\mathscr{L}_2(\mathcal{X}, \frac{\mathbb{P}+\mathbb{Q}}{2})} = 1\}$. The fisher IPM defined on an arbitrary function space $\mathscr{H}$ $d_{\mathscr{H}}(\mathbb{P}, \mathbb{Q})$, approximates the Chi-squared distance. The approximation*

*quality depends on the cosine of the approximation of the optimal critic $f_\chi$ in $\mathscr{H}$. Since $\mathscr{H}$ is symmetric this cosine is always positive (otherwise the same equality holds with an absolute value)*

$$d_{\mathscr{H}}(\mathbb{P},\mathbb{Q}) = \chi_2(\mathbb{P},\mathbb{Q}) \sup_{f \in \mathscr{H} \cap \, \mathbb{S}_{\mathscr{L}_2(\mathcal{X}, \frac{\mathbb{P}+\mathbb{Q}}{2})}} \langle f, f_\chi \rangle_{\mathscr{L}_2(\mathcal{X}, \frac{\mathbb{P}+\mathbb{Q}}{2})},$$

*Equivalently we have following relative approximation error:*

$$\frac{\chi_2(\mathbb{P},\mathbb{Q}) - d_{\mathscr{H}}(\mathbb{P},\mathbb{Q})}{\chi_2(\mathbb{P},\mathbb{Q})} = \frac{1}{2} \inf_{f \in \mathscr{H} \cap \, \mathbb{S}_{\mathscr{L}_2(\mathcal{X}, \frac{\mathbb{P}+\mathbb{Q}}{2})}} \|f - f_\chi\|^2_{\mathscr{L}_2(\mathcal{X}, \frac{\mathbb{P}+\mathbb{Q}}{2})}.$$

From Theorem 2, we know that we have always $d_{\mathscr{H}}(\mathbb{P},\mathbb{Q}) \leq \chi_2(\mathbb{P},\mathbb{Q})$. Moreover if the space $\mathscr{H}$ was rich enough to provide a good approximation of the optimal critic $f_\chi$, then $d_{\mathscr{H}}$ is a good approximation of the Chi-squared distance $\chi_2$.

Generalization bounds for the sample quality of the estimated Fisher IPM from samples from $\mathbb{P}$ and $\mathbb{Q}$ can be done akin to [11], with the main difficulty that for Fisher IPM we have to bound the excess risk of a cost function with data dependent constraints on the function class. We give generalization bounds for learning the Fisher IPM in the supplementary material (**Theorem 3**, Appendix E). In a nutshell the generalization error of the critic learned in a hypothesis class $\mathscr{H}$ from samples of $\mathbb{P}$ and $\mathbb{Q}$, decomposes to the approximation error from Theorem 2 and a statistical error that is bounded using data dependent local Rademacher complexities [17] and scales like $O(\sqrt{1/n}), n = {}^{MN}\!/_{M+N}$. We illustrate in Figure 2 our main theoretical claims on a toy problem.

## 4 Fisher GAN Algorithm using ALM

For any choice of the parametric function class $\mathscr{F}_p$ (for example $\mathscr{F}_{v,\omega}$), note the constraint in Equation (4) by $\hat{\Omega}(f_p, g_\theta) = \frac{1}{2N}\sum_{i=1}^{N} f_p^2(x_i) + \frac{1}{2N}\sum_{j=1}^{N} f_p^2(g_\theta(z_j))$. Define the Augmented Lagrangian [18] corresponding to Fisher GAN objective and constraint given in Equation (4):

$$\mathcal{L}_F(p,\theta,\lambda) = \hat{\mathscr{E}}(f_p, g_\theta) + \lambda(1 - \hat{\Omega}(f_p, g_\theta)) - \frac{\rho}{2}(\hat{\Omega}(f_p, g_\theta) - 1)^2 \qquad (9)$$

where $\lambda$ is the Lagrange multiplier and $\rho > 0$ is the quadratic penalty weight. We alternate between optimizing the critic and the generator. Similarly to [7] we impose the constraint when training the critic only. Given $\theta$, for training the critic we solve $\max_p \min_\lambda \mathcal{L}_F(p,\theta,\lambda)$. Then given the critic parameters $p$ we optimize the generator weights $\theta$ to minimize the objective $\min_\theta \hat{\mathscr{E}}(f_p, g_\theta)$. We give in Algorithm 1, an algorithm for Fisher GAN, note that we use ADAM [19] for optimizing the parameters of the critic and the generator. We use SGD for the Lagrange multiplier with learning rate $\rho$ following practices in Augmented Lagrangian [18].

---

**Algorithm 1** Fisher GAN

---

**Input:** $\rho$ penalty weight, $\eta$ Learning rate, $n_c$ number of iterations for training the critic, N batch size
**Initialize** $p, \theta, \lambda = 0$
**repeat**
    **for** $j = 1$ **to** $n_c$ **do**
        Sample a minibatch $x_i, i = 1 \ldots N, x_i \sim \mathbb{P}_r$
        Sample a minibatch $z_i, i = 1 \ldots N, z_i \sim p_z$
        $(g_p, g_\lambda) \leftarrow (\nabla_p \mathcal{L}_F, \nabla_\lambda \mathcal{L}_F)(p, \theta, \lambda)$
        $p \leftarrow p + \eta$ ADAM $(p, g_p)$
        $\lambda \leftarrow \lambda - \rho g_\lambda$ {SGD rule on $\lambda$ with learning rate $\rho$}
    **end for**
    Sample $z_i, i = 1 \ldots N, z_i \sim p_z$
    $d_\theta \leftarrow \nabla_\theta \hat{\mathscr{E}}(f_p, g_\theta) = -\nabla_\theta \frac{1}{N} \sum_{i=1}^{N} f_p(g_\theta(z_i))$
    $\theta \leftarrow \theta - \eta$ ADAM $(\theta, d_\theta)$
**until** $\theta$ converges

---

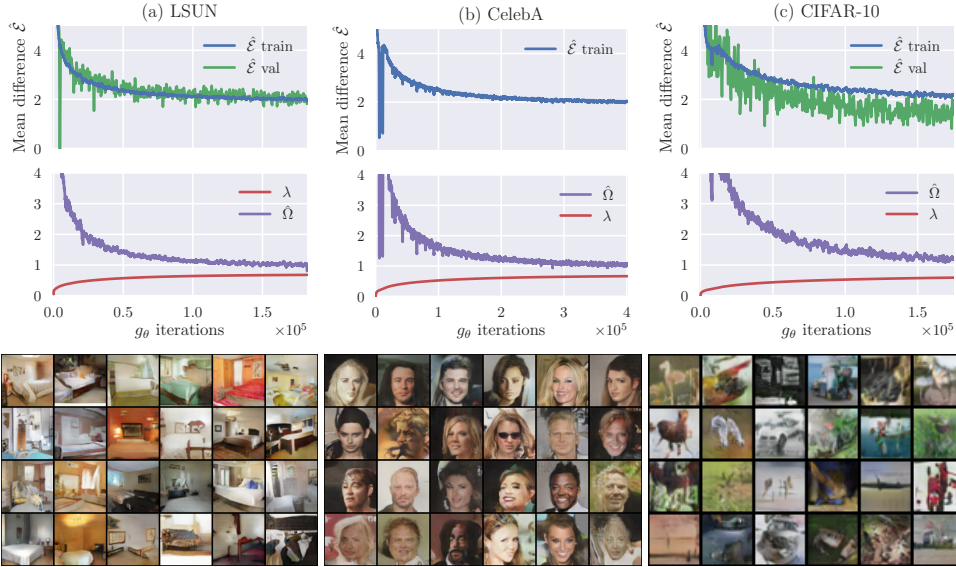

Figure 3: Samples and plots of the loss $\hat{\hat{\mathscr{E}}}(.)$, lagrange multiplier $\lambda$, and constraint $\hat{\Omega}(.)$ on 3 benchmark datasets. We see that during training as $\lambda$ grows slowly, the constraint becomes tight.

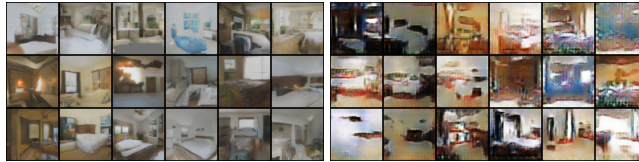

Figure 4: **No Batch Norm:** Training results from a critic $f$ without batch normalization. Fisher GAN (left) produces decent samples, while WGAN with weight clipping (right) does not. We hypothesize that this is due to the implicit whitening that Fisher GAN provides. (Note that WGAN-GP does also succesfully converge without BN [7]). For both models the learning rate was appropriately reduced.

## 5 Experiments

We experimentally validate the proposed Fisher GAN. We claim three main results: (1) stable training with a meaningful and stable loss going down as training progresses and correlating with sample quality, similar to [6, 7]. (2) very fast convergence to good sample quality as measured by inception score. (3) competitive semi-supervised learning performance, on par with literature baselines, without requiring normalization of the critic.

We report results on three benchmark datasets: CIFAR-10 [20], LSUN [21] and CelebA [22]. We parametrize the generator $g_\theta$ and critic $f$ with convolutional neural networks following the model design from DCGAN [23]. For $64 \times 64$ images (LSUN, CelebA) we use the model architecture in Appendix F.2, for CIFAR-10 we train at a $32 \times 32$ resolution using architecture in F.3 for experiments regarding sample quality (inception score), while for semi-supervised learning we use a better regularized discriminator similar to the Openai [9] and ALI [24] architectures, as given in F.4.We used Adam [19] as optimizer for all our experiments, hyper-parameters given in Appendix F.

**Qualitative: Loss stability and sample quality.** Figure 3 shows samples and plots during training. For LSUN we use a higher number of D updates ($n_c = 5$), since we see similarly to WGAN that the loss shows large fluctuations with lower $n_c$ values. For CIFAR-10 and CelebA we use reduced $n_c = 2$ with no negative impact on loss stability. CIFAR-10 here was trained without any label information. We show both train and validation loss on LSUN and CIFAR-10 showing, as can be expected, no overfitting on the large LSUN dataset and some overfitting on the small CIFAR-10 dataset. To back up our claim that Fisher GAN provides stable training, we trained both a Fisher Gan and WGAN where the batch normalization in the critic $f$ was removed (Figure 4).

**Quantitative analysis: Inception Score and Speed.** It is agreed upon that evaluating generative models is hard [25]. We follow the literature in using "inception score" [9] as a metric for the quality

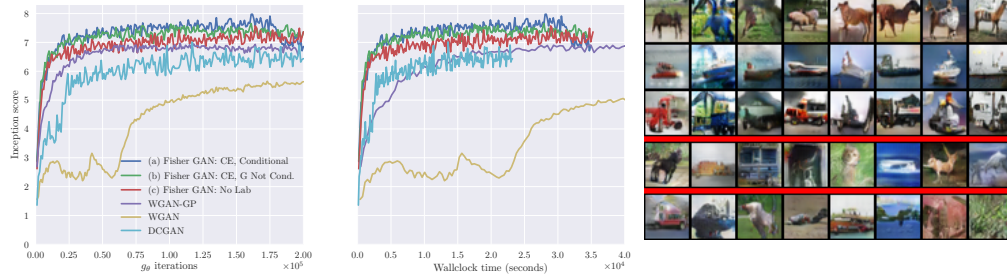

Figure 5: CIFAR-10 inception scores under 3 training conditions. Corresponding samples are given in rows from top to bottom (a,b,c). The inception score plots are mirroring Figure 3 from [7].
**Note** All inception scores are computed from the same tensorflow codebase, using the architecture described in appendix F.3, and with weight initialization from a normal distribution with stdev=0.02. In Appendix F.1 we show that these choices are also benefiting our WGAN-GP baseline.

of CIFAR-10 samples. Figure 5 shows the inception score as a function of number of $g_\theta$ updates and wallclock time. All timings are obtained by running on a single K40 GPU on the same cluster. We see from Figure 5, that Fisher GAN both produces better inception scores, and has a clear speed advantage over WGAN-GP.

**Quantitative analysis: SSL.** One of the main premises of unsupervised learning, is to learn features on a large corpus of unlabeled data in an unsupervised fashion, which are then transferable to other tasks. This provides a proper framework to measure the performance of our algorithm. This leads us to quantify the performance of Fisher GAN by semi-supervised learning (SSL) experiments on CIFAR-10. We do joint supervised and unsupervised training on CIFAR-10, by adding a cross-entropy term to the IPM objective, in conditional and unconditional generation.

Table 2: CIFAR-10 inception scores using resnet architecture and codebase from [7]. We used Layer Normalization [26] which outperformed unnormalized resnets. Apart from this, no additional hyperparameter tuning was done to get stable training of the resnets.

| Method | Score | Method | Score |
|---|---|---|---|
| ALI [24] | $5.34 \pm .05$ | SteinGan [31] | 6.35 |
| BEGAN [27] | 5.62 | DCGAN (with labels, in [31]) | 6.58 |
| DCGAN [23] (in [28]) | $6.16 \pm .07$ | Improved GAN [9] | $8.09 \pm .07$ |
| Improved GAN (-L+HA) [9] | $6.86 \pm .06$ | *Fisher GAN ResNet (ours)* | $8.16 \pm .12$ |
| EGAN-Ent-VI [29] | $7.07 \pm .10$ | AC-GAN [32] | $8.25 \pm .07$ |
| DFM [30] | $7.72 \pm .13$ | SGAN-no-joint [28] | $8.37 \pm .08$ |
| WGAN-GP ResNet [7] | $7.86 \pm .07$ | WGAN-GP ResNet [7] | $8.42 \pm .10$ |
| *Fisher GAN ResNet (ours)* | $7.90 \pm .05$ | **SGAN [28]** | $8.59 \pm .12$ |
| Unsupervised | | Supervised | |

**Unconditional Generation with CE Regularization.** We parametrize the critic $f$ as in $\mathscr{F}_{v,\omega}$. While training the critic using the Fisher GAN objective $\mathcal{L}_F$ given in Equation (9), we train a linear classifier on the feature space $\Phi_\omega$ of the critic, whenever labels are available ($K$ labels). The linear classifier is trained with Cross-Entropy (CE) minimization. Then the critic loss becomes $\mathcal{L}_D = \mathcal{L}_F - \lambda_D \sum_{(x,y) \in \text{lab}} CE(x, y; S, \Phi_\omega)$, where $CE(x, y; S, \Phi_\omega) = -\log \left[ \text{Softmax}(\langle S, \Phi_\omega(\text{x}) \rangle)_\text{y} \right]$, where $S \in \mathbb{R}^{K \times m}$ is the linear classifier and $\langle S, \Phi_\omega \rangle \in \mathbb{R}^K$ with slight abuse of notation. $\lambda_D$ is the regularization hyper-parameter. We now sample three minibatches for each critic update: one labeled batch from the small labeled dataset for the CE term, and an unlabeled batch + generated batch for the IPM.

**Conditional Generation with CE Regularization.** We also trained *conditional generator* models, conditioning the generator on $y$ by concatenating the input noise with a 1-of-K embedding of the label: we now have $g_\theta(z, y)$. We parametrize the critic in $\mathscr{F}_{v,\omega}$ and modify the critic objective as above. We also add a cross-entropy term for the generator to minimize during its training step: $\mathcal{L}_G = \hat{\mathcal{E}} + \lambda_G \sum_{z \sim p(z), y \sim p(y)} CE(g_\theta(z, y), y; S, \Phi_\omega)$. For generator updates we still need to sample only a single minibatch since we use the minibatch of samples from $g_\theta(z, y)$ to compute both the

IPM loss $\hat{\mathscr{E}}$ and CE. The labels are sampled according to the prior $y \sim p(y)$, which defaults to the discrete uniform prior when there is no class imbalance. We found $\lambda_D = \lambda_G = 0.1$ to be optimal.

**New Parametrization of the Critic: "$K + 1$ SSL".** One specific successful formulation of SSL in the standard GAN framework was provided in [9], where the discriminator classifies samples into $K + 1$ categories: the $K$ correct clases, and $K + 1$ for fake samples. Intuitively this puts the real classes in competition with the fake class. In order to implement this idea in the Fisher framework, we define a new function class of the critic that puts in competition the $K$ class directions of the classifier $S_y$, and another "K+1" direction $v$ that indicates fake samples. Hence we propose the following parametrization for the critic: $f(x) = \sum_{y=1}^{K} p(y|x) \langle S_y, \Phi_\omega(x) \rangle - \langle v, \Phi_\omega(x) \rangle$, where $p(y|x) = \text{Softmax}(\langle \text{S}, \Phi_\omega(\text{x}) \rangle)_y$ which is also optimized with Cross-Entropy. Note that this critic does not fall under the interpretation with whitened means from Section 2.2, but does fall under the general Fisher IPM framework from Section 2.1. We can use this critic with both conditional and unconditional generation in the same way as described above. In this setting we found $\lambda_D = 1.5$, $\lambda_G = 0.1$ to be optimal.

**Layerwise normalization on the critic.** For most GAN formulations following DCGAN design principles, batch normalization (BN) [33] in the critic is an essential ingredient. From our semi-supervised learning experiments however, it appears that batch normalization gives substantially worse performance than layer normalization (LN) [26] or even no layerwise normalization. We attribute this to the implicit whitening Fisher GAN provides.

Table 3 shows the SSL results on CIFAR-10. We show that Fisher GAN has competitive results, on par with state of the art literature baselines. When comparing to WGAN with weight clipping, it becomes clear that we recover the lost SSL performance. Results with the $K + 1$ critic are better across the board, proving consistently the advantage of our proposed $K + 1$ formulation. Conditional generation does not provide gains in the setting with layer normalization or without normalization.

Table 3: CIFAR-10 SSL results.

| Number of labeled examples | 1000 | 2000 | 4000 | 8000 |
|---|---|---|---|---|
| Model | | Misclassification rate | | |
| CatGAN [34] | | | 19.58 | |
| Improved GAN (FM) [9] | $21.83 \pm 2.01$ | $19.61 \pm 2.09$ | $18.63 \pm 2.32$ | $17.72 \pm 1.82$ |
| ALI [24] | $19.98 \pm 0.89$ | $19.09 \pm 0.44$ | $17.99 \pm 1.62$ | $17.05 \pm 1.49$ |
| WGAN (weight clipping) Uncond | 69.01 | 56.48 | 40.85 | 30.56 |
| WGAN (weight clipping) Cond | 68.11 | 58.59 | 42.00 | 30.91 |
| Fisher GAN BN Cond | 36.37 | 32.03 | 27.42 | 22.85 |
| Fisher GAN BN Uncond | 36.42 | 33.49 | 27.36 | 22.82 |
| Fisher GAN BN K+1 Cond | 34.94 | 28.04 | 23.85 | 20.75 |
| Fisher GAN BN K+1 Uncond | 33.49 | 28.60 | 24.19 | 21.59 |
| Fisher GAN LN Cond | $26.78 \pm 1.04$ | $23.30 \pm 0.39$ | $20.56 \pm 0.64$ | $18.26 \pm 0.25$ |
| Fisher GAN LN Uncond | $24.39 \pm 1.22$ | $22.69 \pm 1.27$ | $19.53 \pm 0.34$ | $17.84 \pm 0.15$ |
| Fisher GAN LN K+1 Cond | $20.99 \pm 0.66$ | $19.01 \pm 0.21$ | $17.41 \pm 0.38$ | $15.50 \pm 0.41$ |
| Fisher GAN LN K+1, Uncond | $19.74 \pm 0.21$ | $17.87 \pm 0.38$ | $16.13 \pm 0.53$ | $14.81 \pm 0.16$ |
| Fisher GAN No Norm K+1, Uncond | $21.15 \pm 0.54$ | $18.21 \pm 0.30$ | $16.74 \pm 0.19$ | $14.80 \pm 0.15$ |

# 6 Conclusion

We have defined Fisher GAN, which provide a stable and fast way of training GANs. The Fisher GAN is based on a scale invariant IPM, by constraining the second order moments of the critic. We provide an interpretation as whitened (Mahalanobis) mean feature matching and $\chi_2$ distance. We show graceful theoretical and empirical advantages of our proposed Fisher GAN.

**Acknowledgments.** The authors thank Steven J. Rennie for many helpful discussions and Martin Arjovsky for helpful clarifications and pointers.

## Footnotes

[1]Code is available at https://github.com/tomsercu/FisherGAN

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
