[Supplementary Material]

# Supplementary Material for Fisher GAN

Youssef Mroueh*, Tom Sercu*

IBM Research AI

## A WGAN-GP versus Fisher GAN

Consider

$$\mathscr{F}_{v,\omega} = \{f(x) = \langle v, \Phi_w(x) \rangle, v \in \mathbb{R}^m, \Phi_\omega : \mathcal{X} \subset \mathbb{R}^d \to \mathbb{R}^m\}$$

Let

$$J_{\Phi_\omega}(x) \in \mathbb{R}^{m \times d}, [J_{\Phi_\omega}(x)]_{i,j} = \frac{\partial \langle e_i, \Phi_\omega(x) \rangle}{\partial x_j}$$

be the Jacobian matrix of the $\Phi_\omega(.)$. It is easy to see that

$$\nabla_x f(x) = J_{\Phi_\omega}^\top(x) v \in \mathbb{R}^d,$$

and therefore

$$\|\nabla_x f(x)\|^2 = \left\langle v, J_{\Phi_\omega}(x) J_{\Phi_\omega}^\top(x) v \right\rangle,$$

Note that ,

$$J_{\Phi_\omega}(x) J_{\Phi_\omega}^\top(x)$$

is the so called *metric tensor* in information geometry (See for instance [35] and references there in). The gradient penalty for WGAN of [7] can be derived from a Rayleigh quotient principle as well, written in the constraint form:

$$d_{\mathscr{F}_{v,\omega}}(\mathbb{P}, \mathbb{Q}) = \sup_{f \in \mathscr{F}_{v,\omega}, \mathbb{E}_{u \sim U[0,1]} \mathbb{E}_{x \sim u\mathbb{P}+(1-u)\mathbb{Q}} \|\nabla_x f(x)\|^2 = 1} \mathbb{E}_{x \sim \mathbb{P}} f(x) - \mathbb{E}_{x \sim \mathbb{Q}} f(x)$$

Using the special parametrization we can write:

$$\mathbb{E}_{u \sim U[0,1]} \mathbb{E}_{x \sim u\mathbb{P}+(1-u)\mathbb{Q}} \|\nabla_x f(x)\|^2 = v^\top \left( \mathbb{E}_{u \sim U[0,1]} \mathbb{E}_{x \sim u\mathbb{P}+(1-u)\mathbb{Q}} J_{\Phi_\omega}(x) J_{\Phi_\omega}^\top(x) \right) v$$

Let

$$\mathscr{M}_\omega(\mathbb{P}; \mathbb{Q}) = \mathbb{E}_{u \sim U[0,1]} \mathbb{E}_{x \sim u\mathbb{P}+(1-u)\mathbb{Q}} J_{\Phi_\omega}(x) J_{\Phi_\omega}^\top(x) \in \mathbb{R}^{m \times m}$$

is the expected Riemannian metric tensor [35]. Hence we obtain:

$$d_{\mathscr{F}_{v,\omega}}(\mathbb{P}, \mathbb{Q}) = \max_w \max_{v, v^\top \mathscr{M}_\omega(\mathbb{P};\mathbb{Q})v=1} \langle v, \mu_\omega(\mathbb{P}) - \mu_\omega(\mathbb{Q}) \rangle$$

$$= \max_\omega \left\| \mathscr{M}_\omega^{-\frac{1}{2}}(\mathbb{P}; \mathbb{Q})(\mu_\omega(\mathbb{P})) - \mu_\omega(\mathbb{Q}) \right\|$$

Hence Gradient penalty can be seen as well as mean matching in the metric defined by the expected metric tensor $\mathscr{M}_\omega$.

Improved WGAN [7] IPM can be written as follows :

$$\max_\omega \sqrt{(\mu_\omega(\mathbb{P}) - \mu_\omega(\mathbb{Q}))^\top \mathscr{M}_\omega^{-1}(\mathbb{P};\mathbb{Q})(\mu_\omega(\mathbb{P}) - \mu_\omega(\mathbb{Q}))}$$

to be contrasted with Fisher IPM:

$$\max_\omega \sqrt{(\mu_\omega(\mathbb{P}) - \mu_\omega(\mathbb{Q}))^\top \Sigma_\omega^{-1}(\mathbb{P};\mathbb{Q})(\mu_\omega(\mathbb{P}) - \mu_\omega(\mathbb{Q}))}$$

Both Improved WGAN are doing mean matching using different Mahalanobis distances! While improved WGAN uses an *expected metric tensor* $\mathscr{M}_\omega$ to compute this distance, Fisher IPM uses a simple pooled covariance $\Sigma_\omega$ to compute this metric. It is clear that Fisher GAN has a computational advantage!

# B    Chi-squared distance and Pearson Divergence

The definition of $\chi_2$ distance:

$$\chi_2^2(\mathbb{P}, \mathbb{Q}) = 2 \int_{\mathcal{X}} \frac{(\mathbb{P}(x) - \mathbb{Q}(x))^2}{\mathbb{P}(x) + \mathbb{Q}(x)} dx.$$

The $\chi_2$ Pearson divergence:

$$\chi_2^P(\mathbb{P}, \mathbb{Q}) = \int_{\mathcal{X}} \frac{(\mathbb{P}(x) - \mathbb{Q}(x))^2}{\mathbb{Q}(x)} dx.$$

We have the following relation:

$$\chi_2^2(\mathbb{P}, \mathbb{Q}) = \frac{1}{4} \chi_2^P \left( \mathbb{P}, \frac{\mathbb{P} + \mathbb{Q}}{2} \right).$$

# C    Fisher GAN and $\varphi$-divergence Based GANs

Since $f$-gan [3] also introduces a GAN formulation which recovers the Chi-squared divergence, we compare our approaches.

Let us recall here the definition of $\varphi$-divergence:

$$d_\varphi(\mathbb{P}, \mathbb{Q}) = \int_{\mathcal{X}} \varphi \left( \frac{\mathbb{P}(x)}{\mathbb{Q}(x)} \right) \mathbb{Q}(x) dx,$$

where $\varphi : \mathbb{R}^+ \to \mathbb{R}$ is a convex, lower-semicontinuous function satisfying $\varphi(1) = 0$. Let $\varphi^*$ the Fenchel conjugate of $\varphi$:

$$\varphi^*(t) = \sup_{u \in Dom_\varphi} ut - \varphi(u)$$

As shown in [3] and in [36], for any function space $\mathscr{F}$ we get the lower bound:

$$d_\varphi(\mathbb{P}, \mathbb{Q}) \geq \sup_{f \in \mathscr{F}} \mathbb{E}_{x \sim \mathbb{P}} f(x) - \mathbb{E}_{x \sim \mathbb{Q}} \varphi^*(f(x)),$$

For the particular case $\varphi(t) = (t-1)^2$ and $\varphi^*(t) = \frac{1}{4}t^2 + t$ we have the Pearson $\chi_2$ divergence:

$$d_\varphi(\mathbb{P}, \mathbb{Q}) = \int_{\mathcal{X}} \frac{(\mathbb{P}(x) - \mathbb{Q}(x))^2}{\mathbb{Q}(x)} dx = \chi_2^P(\mathbb{P}, \mathbb{Q})$$

Hence to optimize the same cost function of Fisher GAN in the $\varphi$-GAN framework we have to consider:

$$\frac{1}{2} \sqrt{\chi_2^P \left( \mathbb{P}, \frac{\mathbb{P} + \mathbb{Q}}{2} \right)},$$

Fisher GAN gives an inequality for the symmetric Chi-squared and the $\varphi$-GAN gives a lower variational bound. i.e compare for $\varphi$-GAN:

$$\sup_{f \in \mathscr{F}} \mathbb{E}_{x \sim \mathbb{P}} f(x) - \mathbb{E}_{x \sim \frac{\mathbb{P} + \mathbb{Q}}{2}} \varphi^*(f(x)) = \sup_{f \in \mathscr{F}} \mathbb{E}_{x \sim \mathbb{P}} f(x) - \mathbb{E}_{x \sim \frac{\mathbb{P} + \mathbb{Q}}{2}} \left( \frac{1}{4} f^2(x) + f(x) \right)$$

$$= \sup_{f \in \mathscr{F}} \frac{1}{2} \left( \mathbb{E}_{x \sim \mathbb{P}} f(x) - \mathbb{E}_{x \sim \mathbb{Q}} f(x) \right) - \frac{1}{4} \mathbb{E}_{x \sim \frac{\mathbb{P} + \mathbb{Q}}{2}} f^2(x) \quad (10)$$

and for Fisher GAN:

$$\sup_{f \in \mathscr{F}, \mathbb{E}_{x \sim \frac{\mathbb{P} + \mathbb{Q}}{2}} f^2(x) = 1} \mathbb{E}_{x \sim \mathbb{P}} f(x) - \mathbb{E}_{x \sim \mathbb{Q}}(f(x)) \quad (11)$$

while equivalent at the optimum those two formulations for the symmetric Chi-squared given in Equations (10), and (11) have different theoretical and practical properties. On the theory side:

1. While the formulation in (10) is a $\varphi$ divergence, the formulation given by the Fisher criterium in (11) is an IPM with a data dependent constraint. This is a surprising result because $\varphi$-divergences and IPM exhibit different properties and the only known non trivial $\varphi$ divergence that is also an IPM with data independent function class is the total variation distance [36]. When we allow the function class to be dependent on the distributions, the symmetric Chi-squared divergence (in fact general Chi-squared also) can be cast as an IPM! Hence in the context of GAN training we inherit the known stability of IPM based training for GANs.

2. Theorem 2 for the Fisher criterium gives us an approximation error when we change the function from the space of measurable functions to a hypothesis class. It is not clear how tight the lower bound in the $\varphi$-divergence will be as we relax the function class.

On the practical side:

1. Once we parametrize the critic $f$ as a neural network with linear output activation, i.e. $f(x) = \langle v, \Phi_\omega(x) \rangle$, we see that the optimization is unconstrained for the $\varphi$-divergence formulation (10) and the weights updates can explode and have an unstable behavior. On the other hand in the Fisher formulation (11) the data dependent constraint that is imposed slowly through the lagrange multiplier, enforces a variance control that prevents the critic from blowing up and causing instabilities in the training. Note that in the Fisher case we have three players: the critic, the generator and the lagrange multiplier. The lagrange multiplier grows slowly to enforce the constraint and to approach the Chi-squared distance as training converges. Note that the $\varphi$-divergence formulation (10) can be seen as a Fisher GAN with fixed lagrange multiplier $\lambda = \frac{1}{2}$ that is indeed unstable in theory and in our experiments.

**Remark 1.** *Note that if the Neyman divergence is of interest, it can also be obtained as the following Fisher criterium:*

$$\sup_{f \in \mathscr{F}, \mathbb{E}_{x \sim \mathbb{P}} f^2(x) = 1} \mathbb{E}_{x \sim \mathbb{P}} f(x) - \mathbb{E}_{x \sim \mathbb{Q}}(f(x)), \qquad (12)$$

*this is equivalent at the optimum to:*

$$\chi_2^N(\mathbb{P}, \mathbb{Q}) = \int_{\mathcal{X}} \frac{(\mathbb{P}(x) - \mathbb{Q}(x))^2}{\mathbb{P}(x)} dx.$$

*Using a neural network $f(x) = \langle v, \Phi_\omega(x) \rangle$, the Neyman divergence can be achieved with linear output activation and a data dependent constraint:*

$$\sup_{v, \omega, \ \mathbb{E}_{x \sim \mathbb{P}}(\langle v, \Phi_\omega(x) \rangle)^2 = 1} \langle v, \mathbb{E}_{x \sim \mathbb{P}} \Phi_\omega(x) - \mathbb{E}_{x \sim \mathbb{Q}} \Phi_\omega(x) \rangle$$

*To obtain the same divergence as a $\varphi$-divergence we need $\varphi(u) = \frac{(1-u)^2}{u}$, and $\varphi^*(u) = 2 - 2\sqrt{1 - u}$, $(u < 1)$. Moreover exponential activation functions are used in [3], which most likely renders this formulation also unstable for GAN training.*

## D  Proofs

*Proof of Theorem 1.* Consider the space of measurable functions,

$$\mathscr{F} = \left\{ f : \mathcal{X} \to \mathbb{R}, \ f \text{ measurable such that } \int_{\mathcal{X}} f^2(x) \frac{(\mathbb{P}(x) + \mathbb{Q}(x))}{2} dx < \infty \right\}$$

meaning that $f \in \mathscr{L}_2(\mathcal{X}, \frac{\mathbb{P}+\mathbb{Q}}{2})$.

$$
\begin{aligned}
d_{\mathscr{F}}(\mathbb{P}, \mathbb{Q}) &= \sup_{f \in \mathscr{L}_2(\mathcal{X}, \frac{\mathbb{P}+\mathbb{Q}}{2}), f \neq 0} \frac{\mathbb{E}_{x \sim \mathbb{P}}[f(x)] - \mathbb{E}_{x \sim \mathbb{Q}}[f(x)]}{\sqrt{\frac{1}{2}\mathbb{E}_{x \sim \mathbb{P}} f^2(x) + \frac{1}{2}\mathbb{E}_{x \sim \mathbb{Q}} f^2(x)}} \\
&= \sup_{f \in \mathscr{L}_2(\mathcal{X}, \frac{\mathbb{P}+\mathbb{Q}}{2}), \|f\|_{\mathscr{L}_2(\mathcal{X}, \frac{\mathbb{P}+\mathbb{Q}}{2})} = 1} \mathbb{E}_{x \sim \mathbb{P}}[f(x)] - \mathbb{E}_{x \sim \mathbb{Q}}[f(x)] \\
&= \sup_{f \in \mathscr{L}_2(\mathcal{X}, \frac{\mathbb{P}+\mathbb{Q}}{2}), \|f\|_{\mathscr{L}_2(\mathcal{X}, \frac{\mathbb{P}+\mathbb{Q}}{2})} \leq 1} \mathbb{E}_{x \sim \mathbb{P}}[f(x)] - \mathbb{E}_{x \sim \mathbb{Q}}[f(x)] \text{ (By convexity of the cost functional in } f) \\
&= \sup_{f \in \mathscr{L}_2(\mathcal{X}, \frac{\mathbb{P}+\mathbb{Q}}{2})} \inf_{\lambda \geq 0} \mathcal{L}(f, \lambda),
\end{aligned}
$$

where in the last equation we wrote the lagrangian of the Fisher IPM for this particular function class $\mathscr{F} := \mathscr{L}_2(\mathcal{X}, \frac{\mathbb{P}+\mathbb{Q}}{2})$:

$$\mathcal{L}(f, \lambda) = \int_{\mathcal{X}} f(x)(\mathbb{P}(x) - \mathbb{Q}(x))dx + \frac{\lambda}{2}\left(1 - \frac{1}{2}\int_{\mathcal{X}} f^2(x)(\mathbb{P}(x) + \mathbb{Q}(x))dx\right),$$

By convexity of the functional cost and constraints, and since $f \in \mathscr{L}_2(\mathcal{X}, \frac{\mathbb{P}+\mathbb{Q}}{2})$, we can minimize the inner loss to optimize this functional for each $x \in \mathcal{X}$ [37]. The first order conditions of optimality (KKT conditions) gives us for the optimum $f_\chi, \lambda_*$:

$$(\mathbb{P}(x) - \mathbb{Q}(x)) - \frac{\lambda_*}{2}f_\chi(x)(\mathbb{P}(x) + \mathbb{Q}(x))) = 0,$$

$$f_\chi(x) = \frac{2}{\lambda_*}\frac{\mathbb{P}(x) - \mathbb{Q}(x)}{\mathbb{P}(x) + \mathbb{Q}(x)}.$$

Using the feasibility constraint: $\int_{\mathcal{X}} f_\chi^2(x)\left(\frac{\mathbb{P}(x)+\mathbb{Q}(x)}{2}\right) = 1$, we get :

$$\int_{\mathcal{X}} \frac{4}{\lambda_*^2}\frac{(\mathbb{P}(x) - \mathbb{Q}(x))^2}{(\mathbb{P}(x) + \mathbb{Q}(x))^2}\left(\frac{\mathbb{P}(x) + \mathbb{Q}(x)}{2}\right) = 1,$$

which gives us the expression of $\lambda_*$:

$$\lambda_* = \sqrt{\int_{\mathcal{X}} \frac{(\mathbb{P}(x) - \mathbb{Q}(x))^2}{\frac{\mathbb{P}(x)+\mathbb{Q}(x)}{2}}dx}.$$

Hence for $\mathscr{F} := \mathscr{L}_2(\mathcal{X}, \frac{\mathbb{P}+\mathbb{Q}}{2})$ we have:

$$d_{\mathscr{F}}(\mathbb{P}, \mathbb{Q}) = \int_{\mathcal{X}} f_\chi(x)(\mathbb{P}(x) - \mathbb{Q}(x))dx = \sqrt{\int_{\mathcal{X}} \frac{(\mathbb{P}(x) - \mathbb{Q}(x))^2}{\frac{\mathbb{P}(x)+\mathbb{Q}(x)}{2}}dx} = \lambda_*$$

Define the following distance between two distributions:

$$\chi_2(\mathbb{P}, \mathbb{Q}) = \left\|\frac{d\mathbb{P}}{\frac{d\mathbb{P}+d\mathbb{Q}}{2}} - \frac{d\mathbb{Q}}{\frac{d\mathbb{P}+d\mathbb{Q}}{2}}\right\|_{\mathscr{L}_2(\mathcal{X}, \frac{\mathbb{P}+\mathbb{Q}}{2})},$$

We refer to this distance as the $\chi_2$ distance between two distributions. It is easy to see that :

$$d_{\mathscr{F}}(\mathbb{P}, \mathbb{Q}) = \chi_2(\mathbb{P}, \mathbb{Q})$$

and the optimal critic $f_\chi$ has the following expression:

$$f_\chi(x) = \frac{1}{\chi_2(\mathbb{P}, \mathbb{Q})}\frac{\mathbb{P}(x) - \mathbb{Q}(x)}{\frac{\mathbb{P}(x)+\mathbb{Q}(x)}{2}}.$$

$\square$

*Proof of Theorem 2.* Define the means difference functional $\mathscr{E}$:

$$\mathscr{E}(f; \mathbb{P}, \mathbb{Q}) = \mathbb{E}_{x \sim \mathbb{P}} f(x) - \mathbb{E}_{x \sim \mathbb{Q}} f(x)$$

Let

$$\mathbb{S}_{\mathscr{L}_2(\mathcal{X}, \frac{\mathbb{P}+\mathbb{Q}}{2})} = \{f : \mathcal{X} \to \mathbb{R}, \|f\|_{\mathscr{L}_2(\mathcal{X}, \frac{\mathbb{P}+\mathbb{Q}}{2})} = 1\}$$

For a symmetric function class $\mathscr{H}$, the Fisher IPM has the following expression:

$$\begin{aligned} d_{\mathscr{H}}(\mathbb{P}, \mathbb{Q}) &= \sup_{f \in \mathscr{H}, \|f\|_{\mathscr{L}_2(\mathcal{X}, \frac{\mathbb{P}+\mathbb{Q}}{2})}=1} \mathscr{E}(f; \mathbb{P}, \mathbb{Q}) \\ &= \sup_{f \in \mathscr{H} \cap \mathbb{S}_{\mathscr{L}_2(\mathcal{X}, \frac{\mathbb{P}+\mathbb{Q}}{2})}} \mathscr{E}(f; \mathbb{P}, \mathbb{Q}). \end{aligned}$$

Recall that for $\mathscr{H} = \mathscr{L}_2(\mathcal{X}, \frac{\mathbb{P}+\mathbb{Q}}{2})$, the optimum $\chi_2(\mathbb{P}, \mathbb{Q})$ is achieved for :

$$f_\chi(x) = \frac{1}{\chi_2(\mathbb{P}, \mathbb{Q})} \frac{\mathbb{P}(x) - \mathbb{Q}(x)}{\frac{\mathbb{P}(x)+\mathbb{Q}(x)}{2}}, \forall x \in \mathcal{X} \text{ a.s.}$$

Let $f \in \mathscr{H}$ such that $\|f\|_{\mathscr{L}_2(\mathcal{X}, \frac{\mathbb{P}+\mathbb{Q}}{2})} = 1$ we have the following:

$$
\begin{aligned}
\langle f, f_\chi \rangle_{\mathscr{L}_2(\mathcal{X}, \frac{\mathbb{P}+\mathbb{Q}}{2})} &= \int_\mathcal{X} f(x) f_\chi(x) \frac{(\mathbb{P}(x) + \mathbb{Q}(x))}{2} dx \\
&= \frac{1}{\chi_2(\mathbb{P}, \mathbb{Q})} \int_\mathcal{X} f(x)(\mathbb{P}(x) - \mathbb{Q}(x)) dx \\
&= \frac{\mathscr{E}(f; \mathbb{P}, \mathbb{Q})}{\chi_2(\mathbb{P}, \mathbb{Q})}.
\end{aligned}
$$

It follows that for any $f \in \mathscr{H} \cap \mathbb{S}_{\mathscr{L}_2(\mathcal{X}, \frac{\mathbb{P}+\mathbb{Q}}{2})}$ we have:

$$\mathscr{E}(f; \mathbb{P}, \mathbb{Q}) = \chi_2(\mathbb{P}, \mathbb{Q}) \langle f, f_\chi \rangle_{\mathscr{L}_2(\mathcal{X}, \frac{\mathbb{P}+\mathbb{Q}}{2})} \qquad (13)$$

In particular taking the sup over $\mathscr{H} \cap \mathbb{S}_{\mathscr{L}_2(\mathcal{X}, \frac{\mathbb{P}+\mathbb{Q}}{2})}$ we have:

$$d_\mathscr{H}(\mathbb{P}, \mathbb{Q}) = \chi_2(\mathbb{P}, \mathbb{Q}) \sup_{f \in \mathscr{H} \cap \mathbb{S}_{\mathscr{L}_2(\mathcal{X}, \frac{\mathbb{P}+\mathbb{Q}}{2})}} \langle f, f_\chi \rangle_{\mathscr{L}_2(\mathcal{X}, \frac{\mathbb{P}+\mathbb{Q}}{2})}. \qquad (14)$$

note that since $\mathscr{H}$ is symmetric all quantities are positive after taking the sup (if $\mathscr{H}$ was not symmetric one can take the absolute values, and similar results hold with absolute values.)

If $\mathscr{H}$ is rich enough so that we find, for $\varepsilon \in (0, 1)$, a $1 - \varepsilon$ approximation of $f_\chi$ in $\mathscr{H} \cap \mathbb{S}_{\mathscr{L}_2(\mathcal{X}, \frac{\mathbb{P}+\mathbb{Q}}{2})}$, i.e:

$$\sup_{f \in \mathscr{H} \cap \mathbb{S}_{\mathscr{L}_2(\mathcal{X}, \frac{\mathbb{P}+\mathbb{Q}}{2})}} \langle f, f_\chi \rangle_{\mathscr{L}_2(\mathcal{X}, \frac{\mathbb{P}+\mathbb{Q}}{2})} = 1 - \varepsilon$$

we have therefore that $d_\mathscr{H}$ is a $1 - \varepsilon$ approximation of $\chi_2(\mathbb{P}, \mathbb{Q})$:

$$d_\mathscr{H}(\mathbb{P}, \mathbb{Q}) = (1 - \varepsilon)\chi_2(\mathbb{P}, \mathbb{Q}).$$

Since $f$ and $f_\chi$ are unit norm in $\mathscr{L}_2(\mathcal{X}, \frac{\mathbb{P}+\mathbb{Q}}{2})$ we have the following relative error:

$$\frac{\chi_2(\mathbb{P}, \mathbb{Q}) - d_\mathscr{H}(\mathbb{P}, \mathbb{Q})}{\chi_2(\mathbb{P}, \mathbb{Q})} = \frac{1}{2} \inf_{f \in \mathscr{H} \cap \mathbb{S}_{\mathscr{L}_2(\mathcal{X}, \frac{\mathbb{P}+\mathbb{Q}}{2})}} \|f - f_\chi\|^2_{\mathscr{L}_2(\mathcal{X}, \frac{\mathbb{P}+\mathbb{Q}}{2})}. \qquad (15)$$

$\square$

## E  Theorem 3: Generalization Bounds

Let $\mathscr{H}$ be a function space of real valued functions on $\mathcal{X}$. We assume that $\mathscr{H}$ is bounded, there exists $\nu > 0$, such that $\|f\|_\infty \le \nu$. Since the second moments are bounded we can relax this assumption using Chebyshev's inequality, we have:

$$\mathbb{P}\{x \in \mathcal{X}, |f(x)| \le \nu\} \le \frac{\mathbb{E}_{x \sim \frac{\mathbb{P}+\mathbb{Q}}{2}} f^2(x)}{\nu^2} = \frac{1}{\nu^2},$$

hence we have boundedness with high probability. Define the expected mean discrepancy $\mathscr{E}(.)$ and the second order norm $\Omega(.)$:

$$\mathscr{E}(f) = \mathbb{E}_{x \sim \mathbb{P}} f(x) - \mathbb{E}_{x \sim \mathbb{Q}} f(x) \ , \Omega(f) = \frac{1}{2} \left( \mathbb{E}_{x \sim \mathbb{P}} f^2(x) + \mathbb{E}_{x \sim \mathbb{Q}} f^2(x) \right)$$

and their empirical counterparts, given $N$ samples $\{x_i\}_{i=1}^N \sim \mathbb{P}, \{y_i\}_{i=1}^M \sim \mathbb{Q}$ :

$$\hat{\mathscr{E}}(f) = \frac{1}{N} \sum_{i=1}^N f(x_i) - \frac{1}{M} \sum_{i=1}^M f(y_i), \ \hat{\Omega}(f) = \frac{1}{2N} \sum_{i=1}^N f^2(x_i) + \frac{1}{2M} \sum_{i=1}^M f^2(y_i),$$

**Theorem 3.** *Let $n = \frac{MN}{M+N}$. Let $\mathbb{P}, \mathbb{Q} \in \mathscr{P}(\mathcal{X}), \mathbb{P} \neq \mathbb{Q}$, and let $\chi_2(\mathbb{P}, \mathbb{Q})$ be their Chi-squared distance. Let $f^* \in \arg\max_{f \in \mathscr{H}, \Omega(f)=1} \mathscr{E}(f)$, and $\hat{f} \in \arg\max_{f \in \mathscr{H}, \hat{\Omega}(f)=1} \hat{\mathscr{E}}(f)$. Define the expected mean discrepancy of the optimal empirical critic $\hat{f}$:*

$$\hat{d}_{\mathscr{H}}(\mathbb{P}, \mathbb{Q}) = \mathscr{E}(\hat{f})$$

*For $\tau > 0$. The following generalization bound on the estimation of the Chi-squared distance, with probability $1 - 12e^{-\tau}$:*

$$\frac{\chi_2(\mathbb{P}, \mathbb{Q}) - \hat{d}_{\mathscr{H}}(\mathbb{P}, \mathbb{Q})}{\chi_2(\mathbb{P}, \mathbb{Q})} \leq \underbrace{\frac{1}{2} \inf_{f \in \mathscr{H} \cap \mathbb{S}_{\mathscr{L}_2(\mathcal{X}, \frac{\mathbb{P}+\mathbb{Q}}{2})}} \|f - f_\chi\|^2_{\mathscr{L}_2(\mathcal{X}, \frac{\mathbb{P}+\mathbb{Q}}{2})}}_{\text{approximation error}} + \underbrace{\frac{\varepsilon_n}{\chi_2(\mathbb{P}, \mathbb{Q})}}_{\text{Statistical Error}} \quad (16)$$

*where*

$$\varepsilon_n = c_3 \mathscr{R}_{M,N}(f; \{f \in \mathscr{H}, \hat{\Omega}(f) \leq 1 + \nu^2 + 2\eta_n\}, S)$$

$$+ c_4(1 + 2\nu\hat{\lambda})\mathscr{R}_{M,N}(f; \{f \in \mathscr{H}, \hat{\Omega}(f) \leq 1 + \frac{\nu^2}{2} + \eta_n\}, S) + O(\frac{1}{\sqrt{n}})$$

*and*

$$\eta_n \geq c_1 \nu \mathscr{R}_{N,M}(f; f \in \mathscr{H}, S) + c_2 \frac{\nu^2 \tau}{n},$$

*$\hat{\lambda}$ is the Lagrange multiplier, $c_1, c_2, c_3, c_4$ are numerical constants, and $\mathscr{R}_{M,N}$ is the rademacher complexity:*

$$\mathscr{R}_{M,N}(f; \mathscr{F}, S) = E_\sigma \sup_{f \in \mathscr{F}} \left[ \sum_{i=1}^{N+M} \sigma_i \tilde{Y}_i f(X_i) | S \right],$$

*$\tilde{Y} = (\underbrace{\frac{1}{N}, \dots \frac{1}{N}}_{N}, \underbrace{\frac{-1}{M} \dots \frac{-1}{M}}_{M}), S = \{x_1 \dots x_N, y_1 \dots y_M\}, \sigma_i = \pm 1$ with probability $\frac{1}{2}$, that are iids.*

For example:

$$\mathscr{H} = \{f(x) = \langle v, \Phi(x) \rangle, v \in \mathbb{R}^m\}$$

Note that for simplicity here we assume that the feature map is fixed $\Phi : \mathcal{X} \to \mathbb{R}^m$, and we parametrize the class function only with $v$.

$$\mathscr{R}_{M,N}(f; \{\mathscr{H}, \hat{\Omega}(f) \leq R\}, S)) \leq \sqrt{2R \frac{d(\gamma)}{n}},$$

where

$$d(\gamma) = \sum_{j=1}^{m} \frac{\sigma_j^2}{\sigma_j^2 + \gamma}$$

is the effective dimension ($d(\gamma) << m$). Hence we see that typically $\varepsilon_n = O(\frac{1}{\sqrt{n}})$.

*Proof of Theorem 3.* Let $\{x_i\}_{i=1}^{N} \sim \mathbb{P}, \{y_i\}_{i=1}^{M} \sim \mathbb{Q}$. Define the following functionals:

$$\mathscr{E}(f) = \mathbb{E}_{x \sim \mathbb{P}} f(x) - \mathbb{E}_{x \sim \mathbb{Q}} f(x), \Omega(f) = \frac{1}{2}\left(\mathbb{E}_{x \sim \mathbb{P}} f^2(x) + \mathbb{E}_{x \sim \mathbb{Q}} f^2(x)\right)$$

and their empirical estimates:

$$\hat{\mathscr{E}}(f) = \frac{1}{N} \sum_{i=1}^{N} f(x_i) - \frac{1}{M} \sum_{i=1}^{M} f(y_i), \hat{\Omega}(f) = \frac{1}{2N} \sum_{i=1}^{N} f^2(x_i) + \frac{1}{2M} \sum_{i=1}^{M} f^2(y_i)$$

Define the following Lagrangians:

$$\mathcal{L}(f, \lambda) = \mathscr{E}(f) + \frac{\lambda}{2}(1 - \Omega(f)), \hat{\mathcal{L}}(f, \lambda) = \hat{\mathscr{E}}(f) + \frac{\lambda}{2}(1 - \hat{\Omega}(f))$$

Recall some definitions of the Fisher IPM:

$$d_{\mathscr{H}}(\mathbb{P}, \mathbb{Q}) = \sup_{f \in \mathscr{H}} \inf_{\lambda \geq 0} \mathcal{L}(f, \lambda) \text{ achieved at } (f_*, \lambda_*)$$

We assume that a saddle point for this problem exists and it is feasible. We assume also that $\hat{\lambda}$ is positive and bounded.

$$d_{\mathscr{H}}(\mathbb{P}, \mathbb{Q}) = \mathscr{E}(f_*) \text{ and } \Omega(f_*) = 1$$
$$\mathcal{L}(f, \lambda_*) \leq \mathcal{L}(f_*, \lambda_*) \leq \mathcal{L}(f_*, \lambda)$$

The fisher IPM empirical estimate is given by:

$$d_{\mathscr{H}}(\mathbb{P}_N, \mathbb{Q}_N) = \sup_{f \in \mathscr{H}} \inf_{\lambda \geq 0} \hat{\mathcal{L}}(f, \lambda), \text{ achieved at } (\hat{f}, \hat{\lambda})$$

hence we have:

$$d_{\mathscr{H}}(\mathbb{P}_N, \mathbb{Q}_N) = \hat{\mathscr{E}}(\hat{f}) \text{ and } \hat{\Omega}(\hat{f}) = 1.$$

The Generalization error of the empirical critic $\hat{f}$ is the expected mean discrepancy $\mathscr{E}(\hat{f})$. We note $\hat{d}_{\mathscr{H}}(\mathbb{P}, \mathbb{Q}) = \mathscr{E}(\hat{f})$, the estimated distance using the critic $\hat{f}$, on out of samples:

$$\chi_2(\mathbb{P}, \mathbb{Q}) - \hat{d}_{\mathscr{H}}(\mathbb{P}, \mathbb{Q}) = \mathscr{E}(f_\chi) - \mathscr{E}(\hat{f})$$
$$= \underbrace{\mathscr{E}(f_\chi) - \mathscr{E}(f^*)}_{\text{Approximation Error}} + \underbrace{\mathscr{E}(f^*) - \mathscr{E}(\hat{f})}_{\text{Statistical Error}}$$

**Bounding the Approximation Error.** By Theorem 2 we know that:

$$\mathscr{E}(f_\chi) - \mathscr{E}(f^*) = \chi_2(\mathbb{P}, \mathbb{Q}) - d_{\mathscr{H}}(\mathbb{P}, \mathbb{Q}) = \frac{\chi_2(\mathbb{P}, \mathbb{Q})}{2} \inf_{f \in \mathscr{H} \cap \mathbb{S}_{\mathscr{L}_2(\mathcal{X}, \frac{\mathbb{P}+\mathbb{Q}}{2})}} \|f - f_\chi\|^2_{\mathscr{L}_2(\mathcal{X}, \frac{\mathbb{P}+\mathbb{Q}}{2})} .$$

Hence we have for $\mathbb{P} \neq \mathbb{Q}$:

$$\frac{\chi_2(\mathbb{P}, \mathbb{Q}) - \hat{d}_{\mathscr{H}}(\mathbb{P}, \mathbb{Q})}{\chi_2(\mathbb{P}, \mathbb{Q})} = \frac{1}{2} \inf_{f \in \mathscr{H} \cap \mathbb{S}_{\mathscr{L}_2(\mathcal{X}, \frac{\mathbb{P}+\mathbb{Q}}{2})}} \|f - f_\chi\|^2_{\mathscr{L}_2(\mathcal{X}, \frac{\mathbb{P}+\mathbb{Q}}{2})} + \underbrace{\frac{\mathscr{E}(f^*) - \mathscr{E}(\hat{f})}{\chi_2(\mathbb{P}, \mathbb{Q})}}_{\text{Statistical Error}} \qquad (17)$$

Note that this equation tells us that the relative error depends on the approximation error of the the optimal critic $f_\chi$, and the statistical error coming from using finite samples in approximating the distance. We note that the statistical error is divided by the Chi-squared distance, meaning that we need a bigger sample size when $\mathbb{P}$ and $\mathbb{Q}$ are close in the Chi-squared sense, in order to reduce the overall relative error.

Hence we are left with bounding the statistical error using empirical processes theory. Assume $\mathscr{H}$ is a space of bounded functions i.e $\|f\|_\infty \leq \nu$.

**Bounding the Statistical Error.** Note that we have: (i) $\hat{\mathcal{L}}(f^*, \hat{\lambda}) \leq \hat{\mathcal{L}}(\hat{f}, \hat{\lambda})$ and (ii) $\Omega(f^*) = 1$.

$$\mathscr{E}(f^*) - \mathscr{E}(\hat{f}) = \left( \mathscr{E}(f^*) - \hat{\mathscr{E}}(f^*) \right) + (\underbrace{\hat{\mathscr{E}}(f^*) + \frac{\hat{\lambda}}{2}(1 - \hat{\Omega}(f^*)) - \underbrace{\hat{\mathscr{E}}(\hat{f})}_{\hat{\mathcal{L}}(\hat{f}, \hat{\lambda})})}_{\hat{\mathcal{L}}(f^*, \hat{\lambda})} + \left( \hat{\mathscr{E}}(\hat{f}) - \mathscr{E}(\hat{f}) \right) + \frac{\hat{\lambda}}{2} \left( \hat{\Omega}(f^*) - 1 \right)$$

$$\leq \sup_{f \in \mathscr{H}, \Omega(f) \leq 1} |\hat{\mathscr{E}}(f) - \mathscr{E}(f)| + \sup_{f \in \mathscr{H}, \hat{\Omega}(f) \leq 1} |\hat{\mathscr{E}}(f) - \mathscr{E}(f)| + \frac{\hat{\lambda}}{2} \left( \hat{\Omega}(f^*) - \Omega(f^*) \right) \text{ Using (i) and (ii)}$$

$$\leq \sup_{f \in \mathscr{H}, \Omega(f) \leq 1} |\hat{\mathscr{E}}(f) - \mathscr{E}(f)| + \sup_{f \in \mathscr{H}, \hat{\Omega}(f) \leq 1} |\hat{\mathscr{E}}(f) - \mathscr{E}(f)| + \frac{\hat{\lambda}}{2} \sup_{f \in \mathscr{H}, \Omega(f) \leq 1} |\hat{\Omega}(f) - \Omega(f)|.$$

Let $S = \{x_1 \ldots x_N, y_1 \ldots y_M\}$. Define the following quantities:

$$Z_1(S) = \sup_{f \in \mathscr{H}, \Omega(f) \leq 1} |\hat{\mathscr{E}}(f) - \mathscr{E}(f)|, \text{ Concentration of the cost on data distribution dependent constraint}$$

$Z_2(S) = \sup\limits_{f \in \mathcal{H}, \hat{\Omega}(f) \leq 1} |\hat{\mathcal{E}}(f) - \mathcal{E}(f)|$, Concentration of the cost on an empirical data dependent constraint

$Z_3(S) = \sup\limits_{f \in \mathcal{H}, \Omega(f) \leq 1} |\hat{\Omega}(f) - \Omega(f)|$, $\hat{\lambda} Z_3(S)$ is the sensitivity of the cost as the constraint set changes

We have:

$$\mathcal{E}(f^*) - \mathcal{E}(\hat{f}) \leq Z_1(S) + Z_2(S) + \hat{\lambda} Z_3(S), \tag{18}$$

Note that the sup in $Z_1(S)$ and $Z_3(S)$ is taken with respect to class function $\{f, \Omega(f) = \|f\|^2_{\mathcal{L}_2(\mathcal{X}, \frac{\mathbb{P}+\mathbb{Q}}{2})} \leq 1\}$ hence we will bound $Z_1(S)$, and $Z_3(S)$ using local Rademacher complexity. In $Z_2(S)$ the sup is taken on a data dependent function class and can be bounded with local rademacher complexity as well but needs more careful work.

**Bounding $Z_1(S)$, and $Z_3(S)$**

**Lemma 1** (Bounds with (Local) Rademacher Complexity [11, 17]). *Let $Z(S) = \sup_{f \in \mathscr{F}} \mathcal{E}(f) - \hat{\mathcal{E}}(f)$, Assume that $\|f\|_\infty \leq \nu$, for all $f \in \mathscr{F}$.*

- *For any $\alpha, \tau > 0$. Define variances $var_\mathbb{P}(f)$, and similarly $var_\mathbb{Q}(f)$. Assume $\max(var_\mathbb{P}(f), var_\mathbb{Q}(f)) \leq r$ for any $f \in \mathscr{F}$. We have with probability $1 - e^{-\tau}$ :*

$$Z(S) \leq (1 + \alpha) E_S Z(S) + \sqrt{\frac{2r\tau(M+N)}{MN}} + \frac{2\tau\nu(M+N)}{MN}\left(\frac{2}{3} + \frac{1}{\alpha}\right)$$

  *The same result holds for : $Z(S) = \sup_{f \in \mathscr{F}} \hat{\mathcal{E}}(f) - \mathcal{E}(f)$.*

- *By symmetrization we have: $\mathbb{E}_S Z(S) \leq 2 E_S \mathscr{R}_{M,N}(f; \mathscr{F}, S)$ where $\mathscr{R}_{M,N}$ is the rademacher complexity:*

$$\mathscr{R}_{M,N}(f; \mathscr{F}, S) = E_\sigma \sup_{f \in \mathscr{F}} \left[\sum_{i=1}^{N+M} \sigma_i \tilde{Y}_i f(X_i) | S\right],$$

  *$\tilde{Y} = (\underbrace{\frac{1}{N}, \ldots \frac{1}{N}}_{N}, \underbrace{\frac{-1}{M} \ldots \frac{-1}{M}}_{M})$, $\sigma_i = \pm 1$ with probability $\frac{1}{2}$, that are iids.*

- *We have with probability $1 - e^{-\tau}$ for all $\delta \in (0, 1)$:*

$$E_S \mathscr{R}_{M,N}(f; \mathscr{F}, S) \leq \frac{\mathscr{R}_{M,N}(f; \mathscr{F}, S)}{1 - \delta} + \frac{\tau\nu(M+N)}{MN\delta(1-\delta)}.$$

**Lemma 2** (Contraction Lemma [17]). *Let $\phi$ be a contraction, that is $|\phi(x) - \phi(y)| \leq L|x - y|$. Then, for every class $\mathscr{F}$,*

$$\mathscr{R}_{M,N}(f; \phi \circ \mathscr{F}, S) \leq L \mathscr{R}_{M,N}(f; \mathscr{F}, S),$$

$\phi \circ \mathscr{F} = \{\phi \circ f, f \in \mathscr{F}\}$.

Let $n = \frac{MN}{M+N}$. Applying Lemma 1 for $\mathscr{F} = \{f \in \mathcal{H}, \Omega(f) \leq 1\}$. Since $\Omega(f) \leq 1$, $var_\mathbb{P}(f) \leq \Omega(f) \leq 1$, and similarly for $var_\mathbb{Q}(f)$. Hence $\max(var_\mathbb{P}(f), var_\mathbb{Q}(f)) \leq 1$. Putting all together we obtain with probability $1 - 2e^{-\tau}$:

$$Z_1(S) \leq \frac{2(1+\alpha)}{1-\delta}\mathscr{R}_{M,N}(f; \{f \in \mathcal{H}, \Omega(f) \leq 1\}, S) + \sqrt{\frac{2\tau}{n}} + \frac{2\tau\nu}{n}\left(\frac{2}{3} + \frac{1}{\alpha} + \frac{1+\alpha}{\delta(1-\delta)}\right) \tag{19}$$

Now tuning to $Z_3(S)$ applying Lemma 1 for $\{f^2, f \in \mathcal{H}, \Omega(f) \leq 1\}$. Note that $Var(f^2) \leq \mathbb{E}f^4 \leq \Omega(f)\nu^2 \leq \nu^2$. We have that for $\alpha > 0$, $\delta \in (0, 1)$ and with probability at least $1 - 2e^{-\tau}$:

$$Z_3(S) \leq \frac{2(1+\alpha)}{1-\delta}\mathscr{R}_{N,M}(f^2; \{f \in \mathcal{H}, \Omega(f) \leq 1\}, S) + \sqrt{\frac{2\tau\nu^2}{n}} + \frac{2\tau\nu^2}{n}\left(\frac{2}{3} + \frac{1}{\alpha} + \frac{1+\alpha}{\delta(1-\delta)}\right)$$

Note that applying the contraction Lemma for $\phi(x) = x^2$ (with lipchitz constant $2\nu$ on $[-\nu, \nu]$) we have:
$$\mathscr{R}_{N,M}(f^2; \{f \in \mathscr{H}, \Omega(f) \leq 1\}, S) \leq 2\nu\mathscr{R}_{N,M}(f; \{f \in \mathscr{H}, \Omega(f) \leq 1\}, S),$$

Hence we have finally:

$$Z_3(S) \leq \frac{4(1+\alpha)\nu}{1-\delta}\mathscr{R}_{N,M}(f; \{f \in \mathscr{H}, \Omega(f) \leq 1\}, S) + \sqrt{\frac{2\tau\nu^2}{n}} + \frac{2\tau\nu^2}{n}\left(\frac{2}{3} + \frac{1}{\alpha} + \frac{1+\alpha}{\delta(1-\delta)}\right) \tag{20}$$

Note that the of complexity of $\mathscr{H}$, depends also upon the distributions $\mathbb{P}$ and $\mathbb{Q}$, since it is defined on the intersection of $\mathscr{H}$ and the unity ball in $\mathscr{L}_2(\mathcal{X}, \frac{\mathbb{P}+\mathbb{Q}}{2})$.

**From Distributions to Data dependent Bounds.** We study how the $\hat{\Omega}(f)$ concentrates uniformly on $\mathscr{H}$. Note that in this case to apply Lemma 1, we use $r \leq \mathbb{E}(f^4) \leq \nu^4$. We have with probability $1 - 2e^{-\tau}$:

$$\hat{\Omega}(f) \leq \Omega(f) + \frac{4(1+\alpha)\nu}{1-\delta}\mathscr{R}_{N,M}(f; f \in \mathscr{H}, S) + \sqrt{\frac{2\tau\nu^4 r}{n}} + \frac{2\tau\nu^2}{n}\left(\frac{2}{3} + \frac{1}{\alpha} + \frac{1+\alpha}{\delta(1-\delta)}\right)$$

Now using that for any $\alpha > 0$: $2\sqrt{uv} \leq \alpha u + \frac{v}{\alpha}$ we have for $\alpha = \frac{1}{2}$: $\sqrt{\frac{2\tau\nu^4}{n}} \leq \frac{\nu^2}{2} + \frac{4\tau\nu^2}{n}$. For some universal constants, $c_1, c_2$, let:

$$\eta_n \geq c_1\nu\mathscr{R}_{N,M}(f; f \in \mathscr{H}, S) + c_2\frac{\nu^2\tau}{n},$$

we have therefore with probability $1 - 2e^{-\tau}$:

$$\hat{\Omega}(f) \leq \Omega(f) + \frac{\nu^2}{2} + \eta_n, \tag{21}$$

note that Typically $\eta_n = O(\frac{1}{\sqrt{n}})$.
The same inequality holds with the same probability:

$$\Omega(f) \leq \hat{\Omega}(f) + \frac{\nu^2}{2} + \eta_n, \tag{22}$$

Note that we have now the following inclusion using Equation (21):

$$\{f, f \in \mathscr{H}, \Omega(f) \leq 1\} \subset \left\{f, f \in \mathscr{H}, \hat{\Omega}(f) \leq 1 + \frac{\nu^2}{2} + \eta_n\right\}$$

Hence:

$$\mathscr{R}_{M,N}(f; \{f \in \mathscr{H}, \Omega(f) \leq 1\}, S) \leq \mathscr{R}_{M,N}(f; \{f \in \mathscr{H}, \hat{\Omega}(f) \leq 1 + \frac{\nu^2}{2} + \eta_n\}, S)$$

Hence we obtain a data dependent bound in Equations (19),(20) with a union bound with probability $1 - 6e^{-\tau}$.

**Bounding $Z_2(S)$.** Note that concentration inequalities don't apply to $Z_2(S)$ since the cost function and the function class are data dependent. We need to turn the constraint to a data independent constraint i.e does not depend on the training set. For $f, \hat{\Omega}(f) \leq 1$, by Equation (22) we have with probability $1 - 2e^{-\tau}$:

$$\Omega(f) \leq 1 + \frac{\nu^2}{2} + \eta_n,$$

we have therefore the following inclusion with probability $1 - 2e^{-\tau}$:

$$\{f \in \mathscr{H}, \hat{\Omega}(f) \leq 1\} \subset \{f \in \mathscr{H}, \Omega(f) \leq 1 + \frac{\nu^2}{2} + \eta_n\}$$

Recall that:

$$Z_2(S) = \sup_{f \in \mathscr{H}, \hat{\Omega}(f) \leq 1} |\hat{\mathscr{E}}(f) - \mathscr{E}(f)|$$

Hence with probability $1 - 2e^{-\tau}$:

$$Z_2(S) \leq \tilde{Z}_2(S) = \sup_{f, f \in \mathscr{H}, \Omega(f) \leq 1 + \frac{\nu^2}{2} + \eta_n} |\hat{\mathscr{E}}(f) - \mathscr{E}(f)|$$

Applying again Lemma 1 on $\tilde{Z}_2(S)$ we have with probability $1 - 4e^{-\tau}$:

$$Z_2(S) \leq \tilde{Z}_2(S) \leq \frac{2(1 + \alpha)}{1 - \delta} \mathscr{R}_{M,N}(f; \{f \in \mathscr{H}, \Omega(f) \leq 1 + \frac{\nu^2}{2} + \eta_n\}, S) + \sqrt{\frac{2\tau(1 + \frac{\nu^2}{2} + \eta_n)}{n}}$$
$$+ \frac{2\tau\nu}{n}\left(\frac{2}{3} + \frac{1}{\alpha} + \frac{1 + \alpha}{\delta(1 - \delta)}\right).$$

Now reapplying the inclusion using Equation (21), we get the following bound on the local rademacher complexity with probability $1 - 2e^{-\tau}$:

$$\mathscr{R}_{M,N}(f; \{f \in \mathscr{H}, \Omega(f) \leq 1 + \frac{\nu^2}{2} + \eta_n\}, S) \leq \mathscr{R}_{M,N}(f; \{f \in \mathscr{H}, \hat{\Omega}(f) \leq 1 + \nu^2 + 2\eta_n\}, S)$$

Hence with probability $1 - 6e^{-\tau}$ we have:

$$Z_2(S) \leq \frac{2(1 + \alpha)}{1 - \delta} \mathscr{R}_{M,N}(f; \{f \in \mathscr{H}, \hat{\Omega}(f) \leq 1 + \nu^2 + 2\eta_n\}, S) + \sqrt{\frac{2\tau(1 + \frac{\nu^2}{2} + \eta_n)}{n}}$$
$$+ \frac{2\tau\nu}{n}\left(\frac{2}{3} + \frac{1}{\alpha} + \frac{1 + \alpha}{\delta(1 - \delta)}\right).$$

**Putting all together.** We have with probability at least $1 - 12e^{-\tau}$, for universal constants $c_1, c_2, c_3, c_4$

$$\eta_n \geq c_1 \nu \mathscr{R}_{N,M}(f; f \in \mathscr{H}, S) + c_2 \frac{\nu^2 \tau}{n},$$

$$\begin{aligned}
\mathscr{E}(f^*) - \mathscr{E}(\hat{f}) &\leq Z_1(S) + Z_2(S) + \hat{\lambda}Z_3(S) \\
&\leq \varepsilon_n \\
&= c_3 \mathscr{R}_{M,N}(f; \{f \in \mathscr{H}, \hat{\Omega}(f) \leq 1 + \nu^2 + 2\eta_n\}, S) \\
&\quad + c_4(1 + 2\nu\hat{\lambda}) \mathscr{R}_{M,N}(f; \{f \in \mathscr{H}, \hat{\Omega}(f) \leq 1 + \frac{\nu^2}{2} + \eta_n\}, S) \\
&\quad + O(\frac{1}{\sqrt{n}}).
\end{aligned}$$

Note that typically $\varepsilon_n = O(\frac{1}{\sqrt{n}})$. Hence it follows that:

$$\frac{\chi_2(\mathbb{P}, \mathbb{Q}) - \hat{d}_{\mathscr{H}}(\mathbb{P}, \mathbb{Q})}{\chi_2(\mathbb{P}, \mathbb{Q})} \leq \frac{1}{2} \underbrace{\inf_{f \in \mathscr{H} \cap \mathbb{S}_{\mathscr{L}_2(\mathcal{X}, \frac{\mathbb{P}+\mathbb{Q}}{2})}} \|f - f_\chi\|^2_{\mathscr{L}_2(\mathcal{X}, \frac{\mathbb{P}+\mathbb{Q}}{2})}}_{\text{approximation error}} + \underbrace{\frac{\varepsilon_n}{\chi_2(\mathbb{P}, \mathbb{Q})}}_{\text{Statistical Error}}. \quad (23)$$

If $\mathbb{P}$ and $\mathbb{Q}$ are close we need more samples to estimate the $\chi_2$ distance and reduce the relative error.

**Example: Bounding local complexity for a simple linear function class.**

$$\mathscr{H} = \{f(x) = \langle v, \Phi(x) \rangle, v \in \mathbb{R}^m\}$$

Note that for simplicity here we assume that the feature map is fixed $\Phi :$ $\mathcal{X} \rightarrow \mathbb{R}^m$, and we parametrize the class function only with $v$. Note that

$$\sup_{v,v^\top(\Sigma(\mathbb{P}_N)+\Sigma(\mathbb{Q}_M)+\gamma I_m)v\leq 2R}\left\langle v,\sum_{i=1}^{N}\sigma_i\tilde{Y}_i\Phi(X_i)\right\rangle$$

$$=\sup_{v,\|v\|\leq 1}\left\langle v,\left(\frac{\Sigma(\mathbb{P}_N)+\Sigma(\mathbb{Q}_M)+\gamma I_m}{2R}\right)^{-\frac{1}{2}}\sum_{i=1}^{N}\sigma_i\tilde{Y}_i\Phi(X_i)\right\rangle$$

$$=\left\|\left(\frac{\Sigma(\mathbb{P}_N)+\Sigma(\mathbb{Q}_M)+\gamma I_m}{2R}\right)^{-\frac{1}{2}}\sum_{i=1}^{N+M}\sigma_i\tilde{Y}_i\Phi(X_i)\right\|$$

$$=\sqrt{2R}\sqrt{\sum_{i,j=1}^{N+M}\sigma_i\sigma_j\tilde{Y}_i\tilde{Y}_j\Phi(X_i)^\top\left(\Sigma(\mathbb{P}_N)+\Sigma(\mathbb{Q}_M)+\gamma I_m\right)^{-1}\Phi(X_j)}$$

It follows by Jensen inequality that $\mathbb{E}_\sigma \sup_{v,v^\top(\Sigma(\mathbb{P}_N)+\Sigma(\mathbb{Q}_M)+\gamma I_m)v\leq\sqrt{2R}}\left\langle v,\sum_{i=1}^{N}\sigma_i\tilde{Y}_i\Phi(X_i)\right\rangle$

$$\leq\sqrt{2R}\sqrt{\mathbb{E}_\sigma\sum_{i,j=1}^{N+M}\sigma_i\sigma_j\tilde{Y}_i\tilde{Y}_j\Phi(X_i)^\top\left(\Sigma(\mathbb{P}_N)+\Sigma(\mathbb{Q}_M)+\gamma I_m\right)^{-1}\Phi(X_j)}$$

$$=\sqrt{2R}\sqrt{\sum_{i=1}^{N+M}\tilde{Y}_i^2\Phi(X_i)^\top\left(\Sigma(\mathbb{P}_N)+\Sigma(\mathbb{Q}_M)+\gamma I_m\right)^{-1}\Phi(X_i)}$$

$$=\sqrt{2R}\sqrt{Tr\left(\left(\Sigma(\mathbb{P}_N)+\Sigma(\mathbb{Q}_M)+\gamma I_m\right)^{-1}\left(\frac{1}{N}\Sigma(\mathbb{P}_N)+\frac{1}{M}\Sigma(\mathbb{Q}_M)\right)\right)}$$

$$\leq\sqrt{2R\frac{M+N}{MN}}\sqrt{Tr\left(\left(\Sigma(\mathbb{P}_N)+\Sigma(\mathbb{Q}_M)+\gamma I_m\right)^{-1}\left(\Sigma(\mathbb{P}_N)+\Sigma(\mathbb{Q}_M)\right)\right)}$$

Let

$$d(\gamma)=Tr\left(\left(\Sigma(\mathbb{P}_N)+\Sigma(\mathbb{Q}_M)+\gamma I_m\right)^{-1}\left(\Sigma(\mathbb{P}_N)+\Sigma(\mathbb{Q}_M)\right)\right),$$

$d(\gamma)$ is the so called effective dimension in regression problems. Let $\Sigma$ be the singular values of $\Sigma(\mathbb{P}_N)+\Sigma(\mathbb{Q}_M)$,

$$d(\gamma)=\sum_{j=1}^{m}\frac{\sigma_j^2}{\sigma_j^2+\gamma}$$

Hence we obtain the following bound on the local rademacher complexity:

$$\mathscr{R}_{M,N}(f;\{\mathscr{H},\hat{\Omega}(f)\leq R\},S))\leq\sqrt{2R\frac{(M+N)d(\gamma)}{MN}}=\sqrt{2R\frac{d(\gamma)}{n}}$$

Note that without the local constraint the effective dimension $d(\gamma)$ (typically $d(\gamma)<<m$) is replaced by the ambient dimension $m$. $\qquad\square$

# F  Hyper-parameters and Architectures of Discriminator and Generators

For CIFAR-10 we use adam learning rate $\eta=2\mathrm{e}{-4}$, $\beta_1=0.5$ and $\beta_2=0.999$, and penalty weight $\rho=3\mathrm{e}{-7}$, for LSUN and CelebA we use $\eta=5\mathrm{e}{-4}$, $\beta_1=0.5$ and $\beta_2=0.999$, and $\rho=1\mathrm{e}{-6}$. We found the optimization to be stable with very similar performance in the range $\eta\in[1\mathrm{e}{-4},1\mathrm{e}{-3}]$ and $\rho\in[1\mathrm{e}{-7},1\mathrm{e}{-5}]$ across our experiments. We found weight initialization from a normal distribution with stdev=0.02 to perform better than Glorot [38] or He [39] initialization for both Fisher GAN and WGAN-GP. This initialization is the default in pytorch, while in the WGAN-GP codebase He init [39] is used. Specifically the initialization of the generator is more important.

We used some L2 weight decay: $1\mathrm{e}{-6}$ on $\omega$ (i.e. all layers except last) and $1\mathrm{e}{-3}$ weight decay on the last layer $v$.

## F.1 Inception score WGAN-GP baselines: comparison of architecture and weight initialization

As noted in Figure 5 and in above paragraph, we used intialization from a normal distribution with stdev=0.02 for the inception score experiments for both Fisher GAN and WGAN-GP. For transparency, and to show that our architecture and initialization benefits both Fisher GAN and WGAN-GP, we provide plots of different combinations below (Figure 6). Architecture-wise, F64 refers to the architecture described in Appendix F.3 with 64 feature maps after the first convolutional layer. F128 is the architecture from the WGAN-GP codebase [7], which has double the number of feature maps (128 fmaps) and does not have the two extra layers in G and D (D layers 2-7, G layers 9-14). The result reported in the WGAN-GP paper [7] corresponds to `WGAN-GP F128 He init`. For WGAN (Figure 7) the 64-fmap architecture gives some initial instability but catches up to the same level as the 128-fmap architecture.

Figure 6: Architecture and initialization variations, trained with WGAN-GP. Fisher included for comparison. In the main text (Figure 5) we only compare against the best architecture F64 init 0.02.

Figure 7: Architecture variations, trained with WGAN. Fisher included for comparison.

## F.2 LSUN and CelebA.

```
### LSUN and CelebA: 64x64 dcgan with G_extra_layers=2 and
    D_extra_layers=0
G (
  (main): Sequential (
    (0): ConvTranspose2d(100, 512, kernel_size=(4, 4), stride=(1, 1),
        bias=False)
    (1): BatchNorm2d(512, eps=1e-05, momentum=0.1, affine=True)
    (2): ReLU (inplace)
    (3): ConvTranspose2d(512, 256, kernel_size=(4, 4), stride=(2, 2),
        padding=(1, 1), bias=False)
    (4): BatchNorm2d(256, eps=1e-05, momentum=0.1, affine=True)
    (5): ReLU (inplace)
    (6): ConvTranspose2d(256, 128, kernel_size=(4, 4), stride=(2, 2),
        padding=(1, 1), bias=False)
```

```
      (7): BatchNorm2d(128, eps=1e-05, momentum=0.1, affine=True)
      (8): ReLU (inplace)
      (9): ConvTranspose2d(128, 64, kernel_size=(4, 4), stride=(2, 2),
          padding=(1, 1), bias=False)
      (10): BatchNorm2d(64, eps=1e-05, momentum=0.1, affine=True)
      (11): ReLU (inplace)
      (12): Conv2d(64, 64, kernel_size=(3, 3), stride=(1, 1), padding=(1,
          1), bias=False)
      (13): BatchNorm2d(64, eps=1e-05, momentum=0.1, affine=True)
      (14): ReLU (inplace)
      (15): Conv2d(64, 64, kernel_size=(3, 3), stride=(1, 1), padding=(1,
          1), bias=False)
      (16): BatchNorm2d(64, eps=1e-05, momentum=0.1, affine=True)
      (17): ReLU (inplace)
      (18): ConvTranspose2d(64, 3, kernel_size=(4, 4), stride=(2, 2),
          padding=(1, 1), bias=False)
      (19): Tanh ()
  )
)
D (
  (main): Sequential (
    (0): Conv2d(3, 64, kernel_size=(4, 4), stride=(2, 2), padding=(1, 1),
        bias=False)
    (1): LeakyReLU (0.2, inplace)
    (2): Conv2d(64, 128, kernel_size=(4, 4), stride=(2, 2), padding=(1,
        1), bias=False)
    (3): BatchNorm2d(128, eps=1e-05, momentum=0.1, affine=True)
    (4): LeakyReLU (0.2, inplace)
    (5): Conv2d(128, 256, kernel_size=(4, 4), stride=(2, 2), padding=(1,
        1), bias=False)
    (6): BatchNorm2d(256, eps=1e-05, momentum=0.1, affine=True)
    (7): LeakyReLU (0.2, inplace)
    (8): Conv2d(256, 512, kernel_size=(4, 4), stride=(2, 2), padding=(1,
        1), bias=False)
    (9): BatchNorm2d(512, eps=1e-05, momentum=0.1, affine=True)
    (10): LeakyReLU (0.2, inplace)
  )
  (V): Linear (8192 -> 1)
)
```

### F.3  CIFAR-10: Sample Quality and Inceptions Scores Experiments

```
### CIFAR-10: 32x32 dcgan with G_extra_layers=2 and D_extra_layers=2.
    For samples and inception.
G (
  (main): Sequential (
    (0): ConvTranspose2d(100, 256, kernel_size=(4, 4), stride=(1, 1),
        bias=False)
    (1): BatchNorm2d(256, eps=1e-05, momentum=0.1, affine=True)
    (2): ReLU (inplace)
    (3): ConvTranspose2d(256, 128, kernel_size=(4, 4), stride=(2, 2),
        padding=(1, 1), bias=False)
    (4): BatchNorm2d(128, eps=1e-05, momentum=0.1, affine=True)
    (5): ReLU (inplace)
    (6): ConvTranspose2d(128, 64, kernel_size=(4, 4), stride=(2, 2),
        padding=(1, 1), bias=False)
    (7): BatchNorm2d(64, eps=1e-05, momentum=0.1, affine=True)
    (8): ReLU (inplace)
    (9): Conv2d(64, 64, kernel_size=(3, 3), stride=(1, 1), padding=(1,
        1), bias=False)
    (10): BatchNorm2d(64, eps=1e-05, momentum=0.1, affine=True)
    (11): ReLU (inplace)
```

```
    (12): Conv2d(64, 64, kernel_size=(3, 3), stride=(1, 1), padding=(1,
        1), bias=False)
    (13): BatchNorm2d(64, eps=1e-05, momentum=0.1, affine=True)
    (14): ReLU (inplace)
    (15): ConvTranspose2d(64, 3, kernel_size=(4, 4), stride=(2, 2),
        padding=(1, 1), bias=False)
    (16): Tanh ()
  )
)
D (
  (main): Sequential (
    (0): Conv2d(3, 64, kernel_size=(4, 4), stride=(2, 2), padding=(1, 1),
        bias=False)
    (1): LeakyReLU (0.2, inplace)
    (2): Conv2d(64, 64, kernel_size=(3, 3), stride=(1, 1), padding=(1,
        1), bias=False)
    (3): BatchNorm2d(64, eps=1e-05, momentum=0.1, affine=True)
    (4): LeakyReLU (0.2, inplace)
    (5): Conv2d(64, 64, kernel_size=(3, 3), stride=(1, 1), padding=(1,
        1), bias=False)
    (6): BatchNorm2d(64, eps=1e-05, momentum=0.1, affine=True)
    (7): LeakyReLU (0.2, inplace)
    (8): Conv2d(64, 128, kernel_size=(4, 4), stride=(2, 2), padding=(1,
        1), bias=False)
    (9): BatchNorm2d(128, eps=1e-05, momentum=0.1, affine=True)
    (10): LeakyReLU (0.2, inplace)
    (11): Conv2d(128, 256, kernel_size=(4, 4), stride=(2, 2), padding=(1,
        1), bias=False)
    (12): BatchNorm2d(256, eps=1e-05, momentum=0.1, affine=True)
    (13): LeakyReLU (0.2, inplace)
  )
  (V): Linear (4096 -> 1)
  (S): Linear (6144 -> 10)
)
```

### F.4   CIFAR-10: SSL Experiments

```
### CIFAR-10: 32x32 D is in the flavor OpenAI Improved GAN, ALI.
G same as above.

D (
  (main): Sequential (
    (0): Dropout (p = 0.2)
    (1): Conv2d(3, 96, kernel_size=(3, 3), stride=(1, 1), padding=(1, 1))
    (2): LeakyReLU (0.2, inplace)
    (3): Conv2d(96, 96, kernel_size=(3, 3), stride=(1, 1), padding=(1,
        1), bias=False)
    (4): BatchNorm2d(96, eps=1e-05, momentum=0.1, affine=True)
    (5): LeakyReLU (0.2, inplace)
    (6): Conv2d(96, 96, kernel_size=(3, 3), stride=(2, 2), padding=(1,
        1), bias=False)
    (7): BatchNorm2d(96, eps=1e-05, momentum=0.1, affine=True)
    (8): LeakyReLU (0.2, inplace)
    (9): Dropout (p = 0.5)
    (10): Conv2d(96, 192, kernel_size=(3, 3), stride=(1, 1), padding=(1,
        1), bias=False)
    (11): BatchNorm2d(192, eps=1e-05, momentum=0.1, affine=True)
    (12): LeakyReLU (0.2, inplace)
    (13): Conv2d(192, 192, kernel_size=(3, 3), stride=(1, 1), padding=(1,
        1), bias=False)
    (14): BatchNorm2d(192, eps=1e-05, momentum=0.1, affine=True)
    (15): LeakyReLU (0.2, inplace)
```

```
   (16): Conv2d(192, 192, kernel_size=(3, 3), stride=(2, 2), padding=(1,
       1), bias=False)
   (17): BatchNorm2d(192, eps=1e-05, momentum=0.1, affine=True)
   (18): LeakyReLU (0.2, inplace)
   (19): Dropout (p = 0.5)
   (20): Conv2d(192, 384, kernel_size=(3, 3), stride=(1, 1), bias=False)
   (21): BatchNorm2d(384, eps=1e-05, momentum=0.1, affine=True)
   (22): LeakyReLU (0.2, inplace)
   (23): Dropout (p = 0.5)
   (24): Conv2d(384, 384, kernel_size=(3, 3), stride=(1, 1), bias=False)
   (25): BatchNorm2d(384, eps=1e-05, momentum=0.1, affine=True)
   (26): LeakyReLU (0.2, inplace)
   (27): Dropout (p = 0.5)
   (28): Conv2d(384, 384, kernel_size=(1, 1), stride=(1, 1), bias=False)
   (29): BatchNorm2d(384, eps=1e-05, momentum=0.1, affine=True)
   (30): LeakyReLU (0.2, inplace)
   (31): Dropout (p = 0.5)
 )
 (V): Linear (6144 -> 1)
 (S): Linear (6144 -> 10)
)
```

# G   Sample implementation in PyTorch

This minimalistic sample code is based on `https://github.com/martinarjovsky/WassersteinGAN` at commit d92c503.

Some elements that could be added are:

- Validation loop

- Monitoring of weights and activations

- Separate weight decay for last layer $v$ (we trained with $1e-3$ weight decay on $v$).

- Adding Cross-Entropy objective and class-conditioned generator.

## G.1   Main loop

First note the essential change in the critic's forward pass definition:

```
-       output = output.mean(0)
-       return output.view(1)
+       return output.view(-1)
```

Then the main training loop becomes:

```
gen_iterations = 0
for epoch in range(opt.niter):
    data_iter = iter(dataloader)
    i = 0
    while i < len(dataloader):
        ############################
        # (1) Update D network
        ############################
        for p in netD.parameters(): # reset requires_grad
            p.requires_grad = True # they are set to False below in netG update

        # train the discriminator Diters times
        if opt.hiDiterStart and (gen_iterations < 25 or gen_iterations % 500 == 0):
            Diters = 100
        else:
            Diters = opt.Diters
        j = 0
        while j < Diters and i < len(dataloader):
            j += 1

            data = data_iter.next()
            i += 1

            # train with real
```

```
            real_cpu, _ = data
            netD.zero_grad()
            batch_size = real_cpu.size(0)

            if opt.cuda:
                real_cpu = real_cpu.cuda()
            input.resize_as_(real_cpu).copy_(real_cpu)
            inputv = Variable(input)

            vphi_real = netD(inputv)

            # train with fake
            noise.resize_(opt.batchSize, nz, 1, 1).normal_(0, 1)
            noisev = Variable(noise, volatile = True) # totally freeze netG
            fake = Variable(netG(noisev).data)
            inputv = fake

            vphi_fake = netD(inputv)
            # NOTE here f = <v,phi> , but with modified f the below two lines are the
            # only ones that need change. E_P and E_Q refer to Expectation over real and fake.
            E_P_f, E_Q_f = vphi_real.mean(), vphi_fake.mean()
            E_P_f2, E_Q_f2 = (vphi_real**2).mean(), (vphi_fake**2).mean()
            constraint = (1 - (0.5*E_P_f2 + 0.5*E_Q_f2))
            # See Equation (9)
            obj_D = E_P_f - E_Q_f + alpha * constraint - opt.rho/2 * constraint**2
            # max_w min_alpha obj_D. Compute negative gradients, apply updates with negative sign.
            obj_D.backward(mone)
            optimizerD.step()
            # artisanal sgd. We minimze alpha so a <- a + lr * (-grad)
            alpha.data += opt.rho * alpha.grad.data
            alpha.grad.data.zero_()

        ############################
        # (2) Update G network
        ############################
        for p in netD.parameters():
            p.requires_grad = False # to avoid computation
        netG.zero_grad()
        # in case our last batch was the tail batch of the dataloader,
        # make sure we feed a full batch of noise
        noise.resize_(opt.batchSize, nz, 1, 1).normal_(0, 1)
        noisev = Variable(noise)
        fake = netG(noisev)
        vphi_fake = netD(fake)
        obj_G = -vphi_fake.mean() # Just minimize mean difference
        obj_G.backward() # G: min_theta
        optimizerG.step()
        gen_iterations += 1
```

## G.2  Full diff from reference

Note that from the arXiv LaTeX source, the file `diff.txt` could be used in combination with `git apply`.

```
diff --git a/main.py b/main.py
index 7c3e638..e0cae42 100644
--- a/main.py
+++ b/main.py
@@ -34,15 +34,17 @@ parser.add_argument('--cuda' , action='store_true', help='enables cuda')
 parser.add_argument('--ngpu' , type=int, default=1, help='number of GPUs to use')
 parser.add_argument('--netG', default='', help="path to netG (to continue training)")
 parser.add_argument('--netD', default='', help="path to netD (to continue training)")
-parser.add_argument('--clamp_lower', type=float, default=-0.01)
-parser.add_argument('--clamp_upper', type=float, default=0.01)
+parser.add_argument('--wdecay', type=float, default=0.000, help='wdecay value for Phi')
 parser.add_argument('--Diters', type=int, default=5, help='number of D iters per each G iter')
+parser.add_argument('--hiDiterStart' , action='store_true', help='do many D iters at start')
 parser.add_argument('--noBN', action='store_true', help='use batchnorm or not (only for DCGAN)')
 parser.add_argument('--mlp_G', action='store_true', help='use MLP for G')
 parser.add_argument('--mlp_D', action='store_true', help='use MLP for D')
-parser.add_argument('--n_extra_layers', type=int, default=0, help='Number of extra layers on gen and disc')
+parser.add_argument('--G_extra_layers', type=int, default=0, help='Number of extra layers on gen and disc')
+parser.add_argument('--D_extra_layers', type=int, default=0, help='Number of extra layers on gen and disc')
 parser.add_argument('--experiment', default=None, help='Where to store samples and models')
 parser.add_argument('--adam', action='store_true', help='Whether to use adam (default is rmsprop)')
+parser.add_argument('--rho', type=float, default=1e-6, help='Weight on the penalty term for (sigmas -1)**2')
 opt = parser.parse_args()
 print(opt)

@@ -60,7 +62,7 @@ cudnn.benchmark = True
 if torch.cuda.is_available() and not opt.cuda:
     print("WARNING: You have a CUDA device, so you should probably run with --cuda")

-if opt.dataset in ['imagenet', 'folder', 'lfw']:
+if opt.dataset in ['imagenet', 'folder', 'lfw', 'celeba']:
     # folder dataset
     dataset = dset.ImageFolder(root=opt.dataroot,
```

```
                      transform=transforms.Compose([
@@ -94,7 +96,6 @@ nz = int(opt.nz)
 ngf = int(opt.ngf)
 ndf = int(opt.ndf)
 nc = int(opt.nc)
-n_extra_layers = int(opt.n_extra_layers)

 # custom weights initialization called on netG and netD
 def weights_init(m):
@@ -106,11 +107,11 @@ def weights_init(m):
        m.bias.data.fill_(0)

 if opt.noBN:
-    netG = dcgan.DCGAN_G_nobn(opt.imageSize, nz, nc, ngf, ngpu, n_extra_layers)
+    netG = dcgan.DCGAN_G_nobn(opt.imageSize, nz, nc, ngf, ngpu, opt.G_extra_layers)
 elif opt.mlp_G:
    netG = mlp.MLP_G(opt.imageSize, nz, nc, ngf, ngpu)
 else:
-    netG = dcgan.DCGAN_G(opt.imageSize, nz, nc, ngf, ngpu, n_extra_layers)
+    netG = dcgan.DCGAN_G(opt.imageSize, nz, nc, ngf, ngpu, opt.G_extra_layers)

 netG.apply(weights_init)
 if opt.netG != '': # load checkpoint if needed
@@ -120,7 +121,7 @@ print(netG)
 if opt.mlp_D:
    netD = mlp.MLP_D(opt.imageSize, nz, nc, ndf, ngpu)
 else:
-    netD = dcgan.DCGAN_D(opt.imageSize, nz, nc, ndf, ngpu, n_extra_layers)
+    netD = dcgan.DCGAN_D(opt.imageSize, nz, nc, ndf, ngpu, opt.D_extra_layers)
    netD.apply(weights_init)

 if opt.netD != '':
@@ -132,6 +133,7 @@ noise = torch.FloatTensor(opt.batchSize, nz, 1, 1)
 fixed_noise = torch.FloatTensor(opt.batchSize, nz, 1, 1).normal_(0, 1)
 one = torch.FloatTensor([1])
 mone = one * -1
+alpha = torch.FloatTensor([0]) # lagrange multipliers

 if opt.cuda:
    netD.cuda()
@@ -139,14 +141,16 @@ if opt.cuda:
    input = input.cuda()
    one, mone = one.cuda(), mone.cuda()
    noise, fixed_noise = noise.cuda(), fixed_noise.cuda()
+    alpha = alpha.cuda()
+alpha = Variable(alpha, requires_grad=True)

 # setup optimizer
 if opt.adam:
-    optimizerD = optim.Adam(netD.parameters(), lr=opt.lrD, betas=(opt.beta1, 0.999))
-    optimizerG = optim.Adam(netG.parameters(), lr=opt.lrG, betas=(opt.beta1, 0.999))
+    optimizerD = optim.Adam(netD.parameters(), lr=opt.lrD, betas=(opt.beta1, 0.999), weight_decay=opt.wdecay)
+    optimizerG = optim.Adam(netG.parameters(), lr=opt.lrG, betas=(opt.beta1, 0.999), weight_decay=opt.wdecay)
 else:
-    optimizerD = optim.RMSprop(netD.parameters(), lr = opt.lrD)
-    optimizerG = optim.RMSprop(netG.parameters(), lr = opt.lrG)
+    optimizerD = optim.RMSprop(netD.parameters(), lr = opt.lrD, weight_decay=opt.wdecay)
+    optimizerG = optim.RMSprop(netG.parameters(), lr = opt.lrG, weight_decay=opt.wdecay)

 gen_iterations = 0
 for epoch in range(opt.niter):
@@ -160,7 +164,7 @@ for epoch in range(opt.niter):
            p.requires_grad = True # they are set to False below in netG update

        # train the discriminator Diters times
-        if gen_iterations < 25 or gen_iterations % 500 == 0:
+        if opt.hiDiterStart and (gen_iterations < 25 or gen_iterations % 500 == 0):
            Diters = 100
        else:
            Diters = opt.Diters
@@ -168,10 +172,6 @@ for epoch in range(opt.niter):
        while j < Diters and i < len(dataloader):
            j += 1

-            # clamp parameters to a cube
-            for p in netD.parameters():
-                p.data.clamp_(opt.clamp_lower, opt.clamp_upper)
-
            data = data_iter.next()
            i += 1

@@ -185,18 +185,28 @@ for epoch in range(opt.niter):
            input.resize_as_(real_cpu).copy_(real_cpu)
            inputv = Variable(input)

-            errD_real = netD(inputv)
-            errD_real.backward(one)
+            vphi_real = netD(inputv)

            # train with fake
            noise.resize_(opt.batchSize, nz, 1, 1).normal_(0, 1)
            noisev = Variable(noise, volatile = True) # totally freeze netG
```

```
            fake = Variable(netG(noisev).data)
            inputv = fake
-           errD_fake = netD(inputv)
-           errD_fake.backward(mone)
-           errD = errD_real - errD_fake
+
+           vphi_fake = netD(inputv)
+           # NOTE here f = <v,phi> , but with modified f the below two lines are the
+           # only ones that need change. E_P and E_Q refer to Expectation over real and fake.
+           E_P_f, E_Q_f = vphi_real.mean(), vphi_fake.mean()
+           E_P_f2, E_Q_f2 = (vphi_real**2).mean(), (vphi_fake**2).mean()
+           constraint = (1 - (0.5*E_P_f2 + 0.5*E_Q_f2))
+           # See Equation (9)
+           obj_D = E_P_f - E_Q_f + alpha * constraint - opt.rho/2 * constraint**2
+           # max_w min_alpha obj_D. Compute negative gradients, apply updates with negative sign.
+           obj_D.backward(mone)
            optimizerD.step()
+           # artisanal sgd. We minimze alpha so a <- a + lr * (-grad)
+           alpha.data += opt.rho * alpha.grad.data
+           alpha.grad.data.zero_()

        ############################
        # (2) Update G network
@@ -209,14 +219,20 @@ for epoch in range(opt.niter):
        noise.resize_(opt.batchSize, nz, 1, 1).normal_(0, 1)
        noisev = Variable(noise)
        fake = netG(noisev)
-       errG = netD(fake)
-       errG.backward(one)
+       vphi_fake = netD(fake)
+       obj_G = -vphi_fake.mean() # Just minimize mean difference
+       obj_G.backward() # G: min_theta
        optimizerG.step()
        gen_iterations += 1

-       print('[%d/%d][%d/%d][%d] Loss_D: %f Loss_G: %f Loss_D_real: %f Loss_D_fake %f'
+       IPM_enum = E_P_f.data[0] - E_Q_f.data[0]
+       IPM_denom = (0.5*E_P_f2.data[0] + 0.5*E_Q_f2.data[0]) ** 0.5
+       IPM_ratio = IPM_enum / IPM_denom
+       print(('[%d/%d][%d/%d][%d] IPM_enum: %.4f IPM_denom: %.4f IPM_ratio: %.4f '
+           'E_P_f: %.4f E_Q_f: %.4f E_P_(f^2): %.4f E_Q_(f^2): %.4f')
            % (epoch, opt.niter, i, len(dataloader), gen_iterations,
-           errD.data[0], errG.data[0], errD_real.data[0], errD_fake.data[0]))
+           IPM_enum, IPM_denom, IPM_ratio,
+           E_P_f.data[0], E_Q_f.data[0], E_P_f2.data[0], E_Q_f2.data[0]))
        if gen_iterations % 500 == 0:
            real_cpu = real_cpu.mul(0.5).add(0.5)
            vutils.save_image(real_cpu, '{0}/real_samples.png'.format(opt.experiment))
diff --git a/models/dcgan.py b/models/dcgan.py
index 1dd8dbf..ea86a94 100644
--- a/models/dcgan.py
+++ b/models/dcgan.py
@@ -48,9 +48,7 @@ class DCGAN_D(nn.Module):
            output = nn.parallel.data_parallel(self.main, input, range(self.ngpu))
        else:
            output = self.main(input)
-
-       output = output.mean(0)
-       return output.view(1)
+       return output.view(-1)

 class DCGAN_G(nn.Module):
    def __init__(self, isize, nz, nc, ngf, ngpu, n_extra_layers=0):
@@ -148,9 +146,7 @@ class DCGAN_D_nobn(nn.Module):
            output = nn.parallel.data_parallel(self.main, input, range(self.ngpu))
        else:
            output = self.main(input)
-
-       output = output.mean(0)
-       return output.view(1)
+       return output.view(-1)

 class DCGAN_G_nobn(nn.Module):
    def __init__(self, isize, nz, nc, ngf, ngpu, n_extra_layers=0):
```