[Reviews · NeurIPS 2017]

Reviewer 1



In this work, authors present a novel framework for training GAN based on Integral Probability Metrics (IPM). The different notions are introduced gradually which makes the article easy to read. Proof are provided in appendix. Experiments are well managed and complete. - Section 2 - Lines 109 110: I don't see the point of fixing M=N. - Section 5 - One points author could have provided more information is a complexity analysis comparing Wasserstein GAN and Fisher GAN. Where comes the speed up of Fisher-GAN ?

Reviewer 2



Overall the paper is mathematically strong and provides a good and well-founded basis for constraining the discriminator in an adversarial framework trained on IPM. The method is indeed simpler than WGAN-gp and provides a way of constraining the discriminator without the sort of clipping in the original WGAN paper. I liked the paper, but I have several concerns: 1) The constraining factors do indeed have computational advantages over WGAP-gp, where you have to compute the norm of the gradients backpropped onto the input space. Instead we only need to compute the covariance, which is in the simple case just the expectation of the square of the outputs. However, looking at this it seems possible that E[f^2] over samples from the generator go to 0 and 1 over the real samples. In this case, it seems like the gradients in the neighborhood of the samples could still go to 0 with this constraint and a neural network discriminator while the distance given the optimal discriminator are non-zero. WGAN-gp, on the other hand, actually uses a combination of generated and real samples in its regularization term, instead of something over the 2 distributions independently. This likely breaks a symmetry implicit in the Fisher IPM-based method that keeps the gradients from vanishing in the neighborhood of the generated samples. Could the authors comment on this? 2) The stability experiments are lacking. Notably, the authors show stability when the discriminator batch norm is off, but I'm fairly certain that turning the discriminator batch norm off is a strong source of instability. Rather, turning the batch norm off on the generator, for instance, is much stronger. It is interesting that WGAN/gp doesn't perform as well, but it's always unclear if this is due to hyperparameter choices. Why not run the same exps as they did in the WGAN to compare? 3) The story assumes a lot of prior knowledge into IPM, and I think there's a bit of a disconnect between the math and the issues with GANs in general. It could be a bit more accessible. Some of the comments below reflect this. Additional comments: Line 21: Aren’t f-divergences generalizations of the KL and LSGAN (Xi-squared) divergences? Line 27: I'm fairly certain (95%) that WGAN-gp does reduce capacity, and this is evident in the reduced inception scores. Line 115: The authors don't do a good job establishing what v is and how it relates to GANs as most people understand them. In the end, it just ends up being a 1-dimensional thing that disappears under the change of variables, but this point would be lost I think on most readers.

Reviewer 3



This paper proposes a new criterion for training a Generative Adversarial Network and shows that this new criterion yields stability benefits. The criterion is related to Fisher discriminant analysis and is essentially a normalized IPM. The authors show that this criterion is equivalent to the symmetric chi-squared divergence. One reason for not being fully enthusiastic about this paper is that the fact that the proposed criterion is equivalent to chi-squared, which is an f-divergence, reduces the novelty of the approach and raises several questions that are not addressed: - is the proposed implementation significantly different than what one would obtain by applying f-GAN (with the appropriate f corresponding to the chi-squared divergence)? if not, then the algorithm is really just f-GAN with chi-squared divergence and the novelty is drastically reduced. If yes, then it would be great to pinpoint the differences and explain why they are important or would significantly address the issues that are inherent to classical f-divergences based GANs in terms of training stability. - how do the results compare if one uses other f-divergences (for example non-symmetric chi-squared but other f-divergences as well)? - because the criterion is an f-divergence, it may suffer from the same issues that were pointed out in the WGAN paper: gradients would vanish for distributions with disjoint support. Is there a reason why chi-squared would not have the same issues as Jensen-Shannon or KL ? Or is it that the proposed implementation, not being exactly written as chi-squared and only equivalent at the optimum in lambda, doesn't suffer from these issues? Regarding the experimental results: The DCGAN baseline for inceptions scores (Figure 4) seem lower than what was reported in earlier papers and the current state of the art is more around 8 (not sure what it was at the time of the writing of this paper though). The SSL results (Table 2) are ok when compared to the weight clipping implementation of WGAN but this is not necessarily the right baseline to compare with, and compared to other algorithms, the results are not competitive. My overall feeling is that this paper presents an interesting connection between the chi-squared divergence and the normalized IPM coefficient which deserves further investigation (especially a better understanding of how is this connected to f-GAN with chi-squared). But the comparison with the known suboptimal implementation of WGAN (with weight clipping) is not so interesting and the results are not really convincing either. So overall I think the paper is below the acceptance bar in its current form.